# *ReFineVLA*: Multimodal Reasoning-Aware Generalist Robotic Policies via Teacher-Guided Fine-Tuning

## Abstract

Vision-Language-Action (VLA) models have gained much attention from the research community thanks to their strength in translating multimodal observations with linguistic instructions into desired robotic actions. Despite their advancements, VLAs often overlook explicit reasoning and learn the functional input-action mappings, omitting crucial logical steps, which are especially pronounced in interpretability and generalization for complex, long-horizon manipulation tasks. In this work, we propose *ReFineVLA*, a multimodal reasoning-aware framework that fine-tunes VLAs with teacher-guided reasons. We first augment robotic datasets with reasoning rationales generated by an expert teacher model, guiding VLA models to learn to reason about their actions. Then, we fine-tune pre-trained VLAs with the reasoning-enriched datasets with *ReFineVLA*, while maintaining the underlying generalization abilities and boosting reasoning capabilities. We also conduct attention map visualization to analyze the alignment among visual observation, linguistic prompts, and to-be-executed actions of *ReFineVLA*, reflecting the model's ability to focus on relevant tasks and actions. Through this additional step, we explore that *ReFineVLA*-trained models exhibit a meaningful agreement between vision-language and action domains, highlighting the enhanced multimodal understanding and generalization. Evaluated across a suite of simulated manipulation benchmarks on SimplerEnv with both WidowX and Google Robot tasks, *ReFineVLA* achieves state-of-the-art performance, with an average 5.0% improvement in success rate over the second-best method on the WidowX benchmark, reaching 47.7% task success. In more visually and contextually diverse scenarios, *ReFineVLA* yields 3.5% and 2.3% gains in variant aggregation (68.8%) and visual matching (76.6%) settings, respectively. Notably, it improves performance by 9.6% on the *Move Near* task and 8.2% on *Open/Close Drawer* in challenging settings. The source code, models, and all datasets are released anonymously in the appendix materials and will be made publicly available.

## 1 Introduction

Robotic policies learned for task-specific applications often struggle to generalize beyond their training data, limiting their effectiveness when faced with novel objects, environments, instructions, and cross-embodiments. Recent progress in vision-language foundation models, such as CLIP (Radford et al., 2021), LLaVA (Liu et al., 2024a), Phi-3-Vision (Abdin et al., 2024). Along with these foundational models, more powerful large language models (LLMs) and Vision-Language Models (VLMs) have showcased remarkable generalization across a broad spectrum of multimodality. In the same scope, vision-Language-Action (VLA) models (Brohan et al., 2022; Wen et al., 2024; Zhen et al., 2024a; Zheng et al., 2024; Zhou et al., 2025; Wu et al., 2024) have also been proposed to aim for the fusion of broad generalization for foundation models with the specific expertise training on large-scale robotic datasets (Chen et al.; Ebert et al., 2021; Zhu et al., 2023b; O'Neill et al., 2023; Jiang et al., 2024; Khazatsky et al., 2024; Collaboration et al., 2023a).

Although VLAs inherit the robustness of VLMs and are trained on large-scale datasets, these models often lack sophisticated and adaptive multimodal reasoning (Kim et al., 2024; Zhao et al., 2025; Zheng et al., 2024;

Qu et al., 2025). They typically learn a direct functional mapping from multimodal observations to actions, without explicit step-by-step reasoning over the action horizon. This limitation becomes especially pronounced under out-of-distribution conditions, where robust performance requires a nuanced understanding of and adaptability to novel and diverse environmental variations. To address this problem, we propose *ReFineVLA* to inject explicit multimodal reasoning into pre-trained VLAs. Our approach leverages an expert teacher to generate detailed natural-language rationales that articulate the sequential logic as physical intelligence behind robotic actions in the context of language-based instructions. In particular, we hypothesize that this reasoning supervision improves the model's understanding of observation–action relationships, thereby improving its performance on complex and compositional manipulation tasks (Lu et al., 2024; Zhang et al., 2025; Zhou et al., 2025).

Our *ReFineVLA* framework fine-tunes a VLA backbone and embeds it with reasoning-augmented trajectories using a multi-objective loss to predict actions and generate multimodal reasoning rationales. By doing this, the learning of policy and explicit reasoning adaptation is jointly optimized rather than mere input–output mappings, preserving the pre-trained generalization while enhancing reasoning capacity. Specifically, we instantiate *ReFineVLA* by fine-tuning a 3.5B-parameter VLA model on $125,000$ annotated manipulation trajectories with multimodal reasoning annotation. To evaluate generalization ability, we test our models on diverse SimplerEnv scenarios for Google and WidowX robots, which closely replicate real-world conditions. While comparing *ReFineVLA*'s performance against the state-of-the-art methods, our model consistently outperforms existing VLA baselines across all embodiments and environments, demonstrating exceptional robustness under environmental variation, reflecting its deeper multimodal understanding and reasoning. To sum up, our key contributions are summarized below:

- **Methodology:** We propose a transfer-based fine-tuning framework that injects explicit multimodal reasoning into pre-trained VLAs via teacher-generated natural-language rationales, aligning policy learning with structured "chain-of-thought" supervision.

- **Dataset & Models:** We curate a $125,000$-trajectory dataset by prompting an expert reasoning teacher to produce step-by-step multimodal reasoning rationales for each demonstration. We fine-tune 3.5B VLA backbones to learn this reasoning-enriched data, improving performance and inference efficiency through empirical validations and experiments.

- **Attention Visualization & Validation:** We validate our method via extensive simulations on WidowX and Google robots across diverse embodiments. We also conduct attention-map visualizations that reveal that the model's focus shifts from narrow action targets to semantically relevant objects and spatial anchors post–ReFineVLA fine-tuning.

## 2 Related Work

**Vision-Language-Action Models:** Building on the success of VLMs in understanding multimodal data (Karamcheti et al., 2023; Gadre et al., 2022; Driess et al., 2023; Du et al., 2023), recent work has extended these models to robotic control, giving rise to VLAs. These models aim to develop generalist robot policies by enabling pre-trained VLMs to output robot actions. Methods like RT-2 (Brohan et al., 2023), RT-2-X (Collaboration et al., 2023a), OpenVLA (Kim et al., 2024), and RoboPoint (Yuan et al., 2024) treat discretized actions as language tokens and fine-tune large VLMs on robot datasets. Grounding a domain-general vision–language backbone in domain-specific constraints through hybrid explicit–implicit representation learning and task-centric adaptation, as presented by Robotic-CLIP (Nguyen et al., 2024), enables robots to inherit broad visual generalization while accurately modeling action-specific dynamics. Other approaches, like MobilityVLA (Chiang et al., 2024), CogACT (Li et al., 2024a), TraceVLA (Zheng et al., 2024), and $\pi_0$(Black et al., 2024), explore reasoning traces and continuous action spaces. Meanwhile, SpatialVLA(Qu et al., 2025) enhances spatial representations in VLA models. Despite their strong task performance and zero-shot generalization, these models typically require complex fine-tuning for new tasks or novel robot configurations. More importantly, their action-focused training objectives often bypass explicit, step-by-step multimodal reasoning, limiting robustness in tasks that demand deeper understanding and planning (Lu et al., 2024).

**Generalist Robot Policies:** Recent advancements in robotic learning have witnessed a significant trend towards developing multi-task "*generalist*" robot policies capable of performing a wide array of tasks across

diverse environments and embodiments, moving beyond task-specific controllers (Reed et al., 2022; Brohan et al., 2022; Haldar & Pinto, 2023; Zhu et al., 2023a). Early efforts often focused on learning language-conditioned visual policies on a single embodiment using pre-trained visual and text encoders (Parisotto et al., 2015; Rusu et al., 2015; Shridhar et al., 2023; Haldar et al., 2024), which limits their adaptability to new robot platforms. More recent research leverages large-scale, cross-embodiment robot datasets (O'Neill et al., 2024) for pre-training generalist policies, facilitating effective fine-tuning to new robot setups (Team et al., 2024; Liu et al., 2024b; Wang et al., 2024). Notable example, Octo (Team et al., 2024), utilizes a flexible transformer architecture to unify embodiments. Diffusion-based generalist models, such as RPT (Liu et al., 2024b) and HPT (Wang et al., 2024), propose modular architectures to align data from heterogeneous embodiments. In this research line, VLAs represent a prominent direction within generalist policies by directly adapting powerful VLMs for action generation. Nevertheless, they lack the step-by-step reasoning of more sophisticated generalist robot policies. Thus, we aim to solve this problem via an expert-based fine-tuning framework that can work along with and complement generating robot policies.

**Chain-of-Thought Reasoning for Robotics:** Chain-of-Thought (CoT) reasoning has gained prominence in LLMs and VLMs, enabling them to break down complex problems into intermediate steps and improve performance on challenging reasoning tasks (Wei et al., 2022; Kojima et al., 2022; Lyu et al., 2023; Chia et al., 2023; Yao et al., 2023). Methods like fine-tuning smaller models on rationales generated by larger "*teacher*" models have shown promise in transferring reasoning abilities efficiently (Wei et al., 2021). CoT has also been explored in the visual domain for tasks like visual question answering and reasoning about future states (Shao et al., 2024; Rose et al., 2023; Hu et al., 2024; Harvey & Wood, 2023). More recently, CoT-inspired ideas have appeared in embodied AI, including generating textual plans (Mu et al., 2024; Michał et al., 2024), generating robotic trajectories (Wen et al., 2023; Lu et al., 2023; Zawalski et al., 2024), or generating intermediate visual states (Ni et al., 2024; Liang et al., 2024). Nevertheless, the explicit application of multimodal reasoning to guide the policy learning process in VLA models for robotic manipulation, by training the VLA to jointly generate actions and explanatory multimodal reasoning rationales derived from a teacher, remains relatively underexplored. Our work, *ReFineVLA*, also bridges this gap by explicitly leveraging multimodal reasoning rationales generated by an expert LLM teacher to guide the fine-tuning of student VLA models, thereby instilling explicit reasoning capabilities directly into the learned robotic policies.

## 3 A Closer Look at Vision-Language-Action with Multimodal Reasoning

**Background:** Traditional approaches to robotic policy learning often rely on datasets of task-specific demonstrations $\mathcal{D} = \{\tau_1, \tau_2, ..., \tau_n\}$, where each trajectory $\tau_i = \{(o_t, s_t, a_t)\}_{t=1}^T$ records expert observations, states, and actions. The introduced methods (Jain et al., 2024; Zhang et al., 2025; Tschannen et al., 2025) commonly employ a visual encoder $\mathcal{F}_\phi$ to extract features $z_i = \mathcal{F}_\phi(o_i)$ from image observations $o_i$, which are then fed into a policy network $\pi_\theta$ to output action distributions $\hat{a} \sim \pi_\theta(\cdot|z, s)$. Training typically involves minimizing the discrepancy between the predicted actions $\hat{a}$ and the expert actions $a$ (Zhang et al., 2024; Zhen et al., 2024b; Xiang et al., 2025). While effective for specific tasks, this paradigm often struggles with generalization to new tasks, environments, or robot embodiments.

**Limitations of VLAs with Multimodal Reasoning Understanding:** Despite their great foundation in VLMs and training on large robot datasets, standard VLAs can exhibit limitations in deep multimodal understanding and reasoning, as the objective is defined to primarily optimize for accurate next-action prediction without any presence of reasoning-based CoT (Ni et al., 2024; Liang et al., 2024; Mu et al., 2024; Michał et al., 2024). Given an observation **o** including both visual observation, liguistic instruction, and an expert action **a**, the VLA training process minimizes the negative log-likelihood of the action tokens to learn the conditional probability $p(\mathbf{a} \mid \mathbf{o})$, where the standard loss formulation denotes as $\mathcal{L}_{\text{action}} = -\log p(\mathbf{a} \mid \mathbf{o})$. In particular, current VLA models primarily learn a direct functional mapping from visual and linguistic inputs to the corresponding action outputs. While this direct mapping is sufficient for tasks where the required action is a simple, reactive response to the immediate observation and instruction, it often fails to infuse the step-by-step multimodal reasoning needed for more complex scenarios or tasks demanding compositional physical logic (Lu et al., 2024; Zhang et al., 2025; Zhou et al., 2025).

This restriction impacts their ability for robust multimodal understanding for reasoning tasks, such as observation-situation analysis (*i.e.*, interpreting the current state based on visual cues), instructions, and

spatial reasoning (*i.e.*, understanding spatial relationships between objects), and task planning (*i.e.*, breaking down a high-level goal into sequential steps). The action predictor, which focuses solely on the final action outcome, might implicitly encourage the model to find correlations sufficient for seen tasks aggressively, but without explicitly learning the underlying causal or sequential logic, leading to a deficit in robust multimodal understanding.

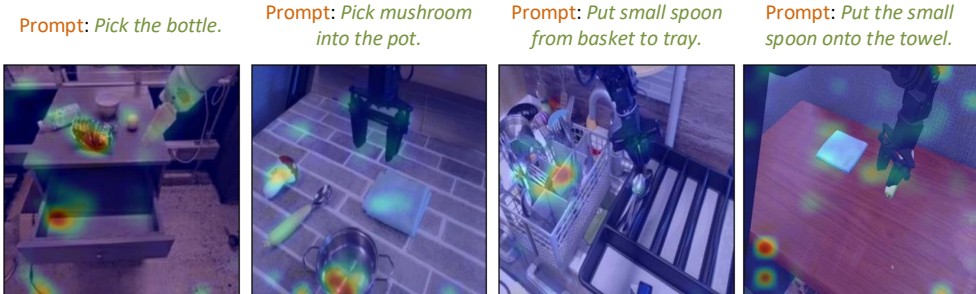

**Figure 1: Visualization of Attention Tokens:** Attention maps of action tokens in standard VLAs, illustrating the narrow focus on visual cues. Darker red shows higher attention scores, while light green means lesser attention.

**To investigate this limitation**, we conduct a closer look at the attention mechanisms within standard VLAs, specifically examining the attention heatmap of action tokens. Motivated by work in visual grounding (Kang et al., 2025), which shows that certain attention heads in VLMs can localize image regions corresponding to text, we analyze where the VLA model attends in the input image when predicting actions. Formally, given an input sequence of visual embeddings and textual prompts, the attention distribution for action tokens during autoregressive decoding is computed as follows:

$$\texttt{Attention}_{h,l}^{(t)}(Q,K,V) = \texttt{softmax}\left(\frac{Q_{h,l}^{(t)}(K_{h,l})^T}{\sqrt{d_k}}\right)V_{h,l}, \tag{1}$$

where $Q_{h,l}^{(t)}$ denotes queries from the $t$-th action token at the $h$-th attention head and the $l$-th layer, $K_{h,l}$ and $V_{h,l}$ represent keys and values derived from the multimodal input sequence, and $d_k$ is the dimension of the key vectors. Subsequently, we aggregate attention scores across heads and layers using a top-$k$ fusion strategy to emphasize the most significant responses based on the attention score in Equation 1:

$$\texttt{AttnMap}^{(t)} = \texttt{top-k}\left(\max_{h\in H, l\in L}\left\{\texttt{Attention}_{h,l}^{(t)}\right\}\right) \tag{2}$$

Via Equation 2, our attention map visualizations in Figure 1 further reveal that traditional VLAs focus narrowly on specific visual cues directly associated with the immediate action, often disregarding crucial broader multimodal context and spatial relationships necessary for deeper reasoning. This narrow focus supports our hypothesis that standard action-only training encourages a reactive, direct mapping rather than a comprehensive multimodal understanding grounded in explicit rationale. We hypothesize that introducing explicit reasoning supervision can mitigate this limitation by enriching the model's understanding of observation–action relationships, thereby enhancing performance on complex and compositional manipulation tasks (Lu et al., 2024; Zhang et al., 2025; Zhou et al., 2025). Further implementation details for attention extraction and visualization can be found in Supplementary Appendix B.6.

## 4  *ReFineVLA*: Reasoning-Aware Teacher-Guided Transfer Fine-Tuning

To enrich VLA models' capability regarding interpretability and generalization, we incorporate *ReFineVLA* with an explicit multimodal reasoning. Specifically, we first present the generation of multimodal reasoning annotations in Section 4.1 from an expert reasoning teacher. Section 4.2 outlines the selective transfer fine-tuning strategy. Thus, the training objectives are formulated in Section 4.3 with the training algorithm (Algorithm 1) and the implementation details (Section 4.4).

Input Prompt: *"You are a robot. Given the image and the instruction to put the spoon into the cup, scan the scene and reason step-by-step about what you see, what each object is, what are the planning steps for a robotic controller, and what actions might be possible? Think carefully and describe the steps followed the key questions."*
 *#1 <Observation>: What do you see in the images?*
 *#2 <Situation Analysis>: What is happening in the scene?*
 *#3 <Spatial Reasoning>: How are the objects arranged, and what spatial relationships matter for completing the task?*
 *#4 <Task Planning>: What are the logical steps to achieve the task, and what should be the robot's sequential actions?*

**Visual Inputs & Linguistic Instruction**

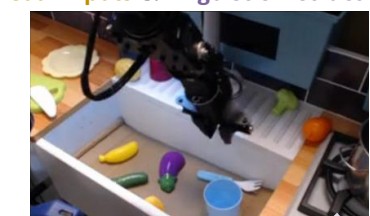

*#1 <Observation>: I observe a toy kitchen sink area. Inside the sink basin, there are several toy items: a yellow banana, a green cucumber, a purple eggplant, a light blue cup, and a light blue spoon with a fork attached. To the right of the sink, on what appears to be a stovetop, there is a metal pot and an orange object (possibly a carrot). Above the sink, there is a robotic arm with a gripper extended downwards, positioned near the back of the sink. There are also other toy items visible on the counter surrounding the sink.*

**Chain-of-Thought Reasoning**

[Δposition, Δrotation, gripper_control]?

*#2 <Situation Analysis>: The scene depicts a toy kitchen environment. The robotic arm is positioned over the sink, which contains several toy items. The task is to put the spoon on the cup.*
*#3 <Spatial Reasoning>: The spoon and the cup are both located inside the sink basin. The cup is positioned upright and closer to the front right corner of the sink from the robot's perspective. The spoon, which appears to be part of a spoon-fork utensil, is lying flat next to the cup, slightly to its left. The robotic arm's gripper is currently positioned above the back of the sink, not directly over the spoon or cup. To complete the task, the robot needs to interact with both the spoon and the cup. The spoon is currently not on the cup.*
*#4 <Task Planning>: The robot has to move its arm to a position suitable for grasping the spoon.*
 *"logical_steps":*
  *- Identify and locate the spoon: The spoon is identifiable and its position in the sink is known.*
  *- Identify and locate the cup: The cup is identifiable and its position in the sink is known.*
  *- Approach the spoon: The robot needs to move its gripper closer to the spoon.*
  *- Grasp the spoon: The robot must successfully pick up the spoon with its gripper.*
  *- Move the spoon towards the cup: Once grasped, the robot needs to lift the spoon and maneuver it over the cup.*
  *- Place the spoon on the cup: The robot must lower the spoon onto the rim or opening of the cup.*
  *- Release the spoon: The robot will then open its gripper to release the spoon onto the cup.*
 *"sub_action": Move its arm to a position suitable for grasping the spoon.*

**Figure 2: Multimodal Robotic Instruction Understanding with Chain-of-Thought Reasoning**: An illustrative example depicts a single annotated data sample from a robotic manipulator grounded task planning. The task is for a robot to place a spoon into a cup, given an observation of a cluttered scene and natural language instructions. The input prompt guides the robot to reason through a sequence of structured questions: (1) *Observation* – identifying objects in the image; (2) *Situation Analysis* – understanding the context; (3) *Spatial Reasoning* – analyzing object relationships; and (4) *Task Planning* – formulating an action plan in logical steps. The annotated response includes step-by-step reasoning under each category, leading to a detailed plan of robot actions involving position, rotation, and gripper control for low-level motor commands.

## 4.1 Multimodal Reasoning Annotation Generation

Standard robotic datasets, denoted as $\mathcal{D} = \{(o_i, a_i)\}_{i=1}^{N}$, contain multimodal observations $o_i$, including visual images $I_i$ and language instructions $l_i$, paired with actions $a_i$. These datasets aim to capture a variety of tasks and scenarios that robots may encounter, facilitating the training of models that map observations to appropriate actions. However, a critical limitation of these datasets is the lack of explicit reasoning annotations that explain *why* specific actions are suitable given the multimodal inputs. In other words, while these datasets provide examples of what a robot should do in a given situation, they do not offer insight into the underlying decision-making process or rationale that justifies each action.

To explicitly incorporate multimodal reasoning, we utilize powerful reasoning-based teacher models (*i.e.*, Gemini (Team et al., 2023)). We generate reasoning annotations by prompting the teacher with the structured

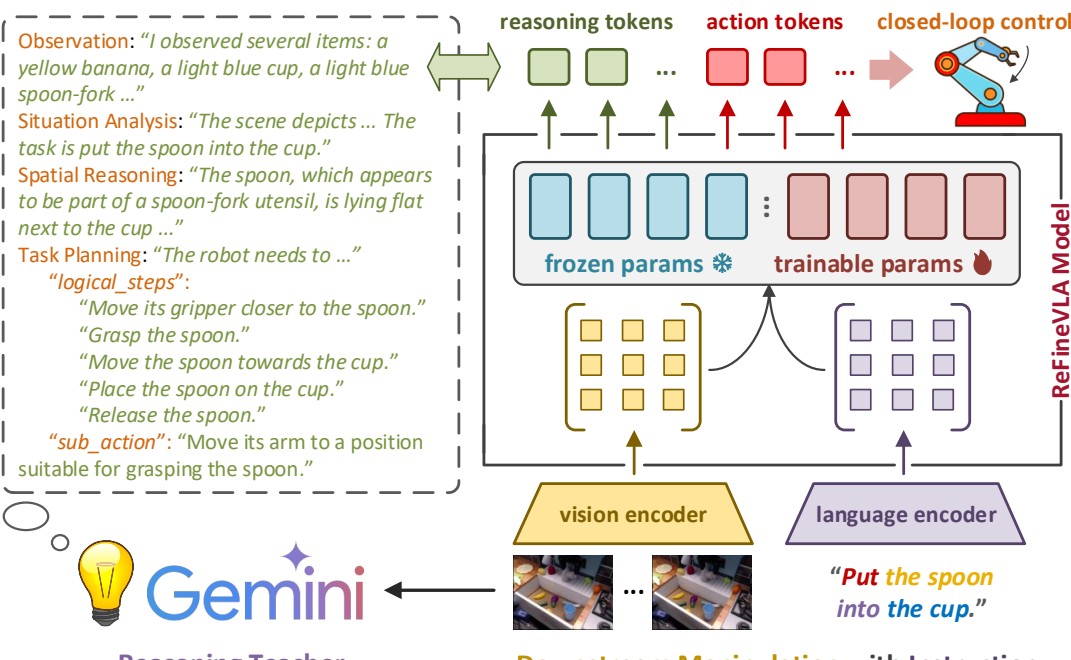

**Figure 3: Overview of *ReFineVLA*'s Training Flow:** A fine-tuning framework that enhances VLA models with explicit multimodal reasoning, guided by rationales from a teacher model. These rationales cover visual cues, spatial reasoning, and task planning and are injected during training via action and reasoning losses. The learner integrates visual-linguistic inputs, infuses reasoning, and outputs interpretable actions for closed-loop control.

multimodal reasoning prompt (Figure 2). For each observation-action pair $(o_i, a_i)$, the teacher model generates a detailed multimodal reasoning annotation $r_i$, explicitly elucidating the rationale behind the chosen action given the visual and linguistic context. The enriched dataset thus becomes $\mathcal{D}' = \{(o_i, a_i, r_i)\}_{i=1}^{N}$ with $r_i$ denotes as the reasoning for $i^{\text{th}}$ pair in the original dataset $\mathcal{D}$. These annotations act as structured multimodal reasoning signals, bridging perception and action. Explicit rationales enable models to understand and reason through complex task requirements, significantly enhancing their ability to generalize across tasks and interpret their decision-making processes. Extended examples similar to Figure 2 are provided in Table 4, Table 5, and Table 6 (Appendix B.1).

## 4.2 Selective Transfer Fine-Tuning for Efficient Adaptation

*ReFineVLA* builds upon robust features learned by large-scale VLA models pre-trained on extensive general robotic manipulation data. Rather than fully fine-tuning the model, which can be computationally expensive, *ReFineVLA* employs selective transfer fine-tuning, outlined as follows:

- **Preservation of General Features:** Pre-trained VLA models encode diverse and generalizable features. Full fine-tuning on specialized reasoning-enriched datasets risks overfitting, thereby losing foundational generalization (Zhou et al., 2025). Therefore, selective fine-tuning preserves these foundational representations by freezing most parameters, particularly the lower layers involved in basic feature extraction (Del Corro et al., 2023; Yue et al., 2024).

- **Efficient Adaptation:** Selectively tuning a subset of model parameters significantly reduces computational resource demands, such as time, memory, and FLOPs, making *ReFineVLA* practical and scalable for diverse robotic tasks and embodiments.

- **Targeted Knowledge Infusion:** Explicit multimodal reasoning predominantly involves higher-level cognitive processing. We hypothesize that such abstract reasoning capabilities reside in the upper layers of the VLA model, typically associated with decision-making and complex feature integration. Selective fine-tuning targets these upper layers, enabling the model to effectively integrate explicit reasoning capabilities without compromising foundational low-level feature extraction.

Empirically determined layers, such as later transformer blocks in vision and language encoders and the policy head, are selected to ensure targeted reasoning infusion, preserving general pre-trained features while efficiently adapting the model for explicit reasoning.

### 4.3 Learning Objectives

To successfully realize the shortcomings in robotic multimodal reasoning as outlined above, we define the training objective of *ReFineVLA* that jointly optimizes action prediction and reasoning generation as follows:

$$\mathcal{L}_{\text{ReFineVLA}} = \mathcal{L}_{\text{action}} + \lambda_{\text{r}} \mathcal{L}_{\text{reasoning}}, \tag{3}$$

where $\mathcal{L}_{\text{action}}$ presents the behavioral cloning loss ensuring accurate action cloning/prediction, $\mathcal{L}_{\text{reasoning}}$ guides rationale generation, fostering structured multimodal reasoning, and $\lambda_{\text{r}}$ serves as a hyperparameter controlling the penalty of *ReFineVLA*'s reasoning performance.

**Action Prediction Loss ($\mathcal{L}_{\textbf{action}}$):** The model predicts ground-truth actions $a_i$ given multimodal inputs $o_i$, optimizing the negative log-likelihood:

$$\mathcal{L}_{\text{action}} = -\sum_t \log \mathbb{P}(a_{i,t} \mid o_i, a_{i,<t}; \theta). \tag{4}$$

**Reasoning Generation Loss ($\mathcal{L}_{\textbf{reasoning}}$):** The model is also trained to produce rationales $r_i$, governed through standard language modeling negative log-likelihood as follows:

$$\mathcal{L}_{\text{reasoning}} = -\sum_j \log \mathbb{P}(r_{i,j} \mid o_i, r_{i,<j}; \theta). \tag{5}$$

In brief. Equation 4 ensures the model learns the primary task as an objective for actions; meanwhile, Equation 5 governs the model to learn thinking step-by-step, akin to a reasoning objective.

---

**Algorithm 1:** *ReFineVLA*: Reasoning-Aware Fine-Tuning of VLA

---

**Input:** Reasoning dataset $\mathcal{D}' = \{(x_i, r_i, a_i)\}_{i=1}^N$; pre-trained parameters $\theta_{\text{full}}$; batch size $B$
**Output:** Fine-tuned subset of parameters $\theta$

**1** Freeze $\theta_{\text{full}} \setminus \theta$                                     `// only fine-tune upper layers`
**2** **repeat**
**3**      Sample mini-batch $\{(o_j, a_j, r_j)\}_{j=1}^B \sim \mathcal{D}'$
**4**      **for** $j = 1$ **to** $B$ **do**
**5**          $\hat{a}_j \leftarrow \text{VLA}_\theta(o_j)$                                 `// action prediction`
**6**          $\hat{r}_j \leftarrow \text{VLA}_\theta(o_j)$                        `// auto-regressive rationale generation`
**7**          $\mathcal{L}_{\text{action}}^{(j)} \leftarrow -\log \mathbb{P}(a_j \mid x_j; \theta)$                 `// action objective (Equation 4)`
**8**          $\mathcal{L}_{\text{reasoning}}^{(j)} \leftarrow -\log \mathbb{P}(r_j \mid x_j; \theta)$           `// reasoning objective (Equation 5)`
**9**      $\mathcal{L}_{\text{batch}} \leftarrow \frac{1}{B} \sum_{b=1}^B \left( \mathcal{L}_{\text{action}}^{(b)} + \lambda_r \mathcal{L}_{\text{reasoning}}^{(b)} \right)$         `// model objective (Equation 3)`
**10**      $\theta \leftarrow \theta - \eta \nabla_\theta \mathcal{L}_{\text{batch}}$                                `// parameters update`
**11** **until** *convergence (e.g., validation performance plateaus or training epochs reached)*

---

### 4.4 Implementation Details

The training of *ReFineVLA* in Algorithm 1 follows a supervised learning paradigm, iteratively updating the selected learnable parameters, $\theta$, of the pre-trained VLA model. Through this, we are able to refine any VLAs to jointly predict actions and generate rationales, leveraging pre-trained knowledge while efficiently adapting to the reasoning-augmented data, as illustrated in Figure 3. For more about data generation and implementation details, please refer to Appendix B.

We apply our multimodal reasoning annotation generation pipeline to the BridgeData-v2 (Walke et al., 2023) and Google RT1 Robot datasets (Brohan et al., 2022) to create our datasets with reasoning annotations.

Specifically, we gathered approximately 125,000 robot trajectories annotated with multimodal reasoning annotation, forming our fine-tuning dataset for *ReFineVLA*. For VLA models, we started with SpatialVLA (Qu et al., 2025), a 3.5B VLA model based on the PaliGemma 2 VLM backbone (Steiner et al., 2024), trained on the Open X-Embodiment (O'Neill et al., 2024) and RHT20 datasets (Fang et al., 2024). *SpatialVLA* is also pre-trained on a large dataset, exploring effective spatial representations through techniques like Ego3D Position Encoding and Adaptive Action Grids, which enhance 3D scene understanding and transferability. These models provide robust starting points with established capabilities in generalist policies and spatial awareness, respectively, making them suitable backbones for learning enhanced reasoning.

## 5 Experiments & Ablation Studies

### 5.1 Baselines

To evaluate our method's performance, we conduct experiments on diverse environments of 137 configurations across Google Robot tasks and WidowX Robot tasks in SimplerEnv that closely mimic real-world settings and conditions. We compare against state-of-the-art open-source generalist VLA-based policies with varying model sizes and training paradigms.

**OpenVLA** (Kim et al., 2024): A VLA model built upon Llama-2 (Touvron et al., 2023) combined with a visual encoder integrating pre-trained features from DINOv2 (Oquab et al., 2023) and SigLIP (Zhai et al., 2023). OpenVLA is pre-trained on the Open-X-Embodiment dataset (Collaboration et al., 2023b).

**Octo-Base** (Team et al., 2024): A-93M parameter transformer-based policy trained on 800,000 trajectories from Open-X-Embodiment.

**RT-1** (O'Neill et al., 2024): A scalable Robotics Transformer model to transfer knowledge from large task-agnostic datasets. Trained on diverse robotic datasets, RT-1 attains a high level of generalization and task-specific performance across a variety of robotic tasks, demonstrating the value of open-ended task-agnostic training of high-capacity models.

**RoboVLM** (Dorka et al., 2024): A unified, flexible framework for converting pre-trained VLM-based models into VLA-based policies by systematically exploring three key design dimensions with VLM backbone selection, policy architecture formulation, such as action-space and history integration, and timing of cross-embodiment.

**TraceVLA** (Zheng et al., 2024): An enhanced spatial-temporal VLA model for reasoning via visual trace prompting. Built by fine-tuning OpenVLA on robot manipulation trajectories, TraceVLA is able to encode state-action history as visual prompts to improve manipulation performance in interactive tasks.

**SpatialVLA** (Qu et al., 2025): The most recent state-of-the-art model focused on spatial understanding for robot manipulation, incorporating 3D information such as spatial movement. SpatialVLA learns a generalist policy for spatial manipulation across diverse robotic tasks and predicts four actions at a time.

### 5.2 Simulation Evaluation

**SimplerEnv:** Our simulation evaluation utilizes SimplerEnv Google Robot tasks, which follow two distinct settings: *visual matching* and *variant aggregation*. The visual matching setting aims to minimize the visual appearance gap between real environments and raw simulation, significantly enhancing the correlation between policy performance in simulation and real-world scenarios. Complementing this, variant aggregation covers a wide range of environmental variations, including backgrounds of different rooms, lighter and darker lighting conditions, varying numbers of distractors, solid color and complex table textures, and camera poses. We also evaluate across different manipulation policies on the SimplerEnv WidowX Robot tasks setup with tasks: *Put Spoon on Towel*, *Put Carrot on Plate*, *Stack Green Block on Yellow Block*, and *Put Eggplant in Yellow Basket*, measuring both grasp correction success and full task completion rates. These variations allow us to assess the robustness and adaptability of *ReFineVLA* in handling diverse manipulation scenarios, particularly in evaluating the spatial and temporal awareness brought by multimodal reasoning understanding.

**Table 1: Evaluated Performances of VLA baselines on SimplerEnv with WidowX Robot Tasks:** The zero-shot and fine-tuning results denote the performance of the pre-trained models on the OXE dataset (O'Neill et al., 2024) and fine-tuned models on the BridgeData-v2 (Walke et al., 2023), respectively. **Bold** denotes the best peformance and underline indicates the second-best one.

| Robotic Task / VLA Baseline | Put Spoon on Towel | | Put Carrot on Plate | | Stack Green Block on Yellow Block | | Put Eggplant in Yellow Basket | | Average |
|---|---|---|---|---|---|---|---|---|---|
| | Grasp | Success | Grasp | Success | Grasp | Success | Grasp | Success | |
| RT-1-X (Collaboration et al., 2023b) | 16.7% | 0.0% | 20.8% | 4.2% | 8.3% | 0.0% | 0.0% | 0.0% | 1.1% |
| Octo-Base (Team et al., 2024) | 34.7% | 12.5% | **52.8%** | 8.3% | 31.9% | 0.0% | 66.7% | 43.1% | 16.0% |
| Octo-Small (Team et al., 2024) | **77.8%** | **47.2%** | 27.8% | 9.7% | 40.3% | 4.2% | 87.5% | 56.9% | 30.0% |
| OpenVLA (Kim et al., 2024) | 4.1% | 0.0% | 33.3% | 0.0% | 12.5% | 0.0% | 8.3% | 4.1% | 1.1% |
| RoboVLM (zero-shot) (Dorka et al., 2024) | 37.5% | 20.8% | 33.3% | 25.0% | 8.3% | 8.3% | 0.0% | 0.0% | 13.5% |
| RoboVLM (fine-tuning) (Dorka et al., 2024) | 54.2% | 29.2% | 25.0% | 25.0% | 45.8% | 12.5% | 58.3% | 58.3% | 31.3% |
| SpatialVLA (zero-shot) (Qu et al., 2025) | 25.0% | 20.8% | 41.7% | 20.8% | 58.3% | 25.0% | 79.2% | 70.8% | 34.4% |
| SpatialVLA (fine-tuning) (Qu et al., 2025) | 20.8% | 16.7% | 29.2% | 25.0% | 62.5% | **29.2%** | **100.0%** | **100.0%** | 42.7% |
| *ReFineVLA* (Ours) | 42.9% | 38.1% | 33.3% | **33.3%** | **71.4%** | 23.8% | **100.0%** | 95.2% | **47.7%** |

**Table 2: Evaluated Performances of VLA baselines on SimplerEnv with Google Robot Tasks:** The zero-shot and fine-tuning results denote the performance of the pre-trained models on the OXE dataset (O'Neill et al., 2024) and the fine-tuned models on the Fractal dataset (Brohan et al., 2022), respectively. **Bold** denotes the best peformance and underline indicates the second-best one.

| Robotic Task / VLA Baseline | Visual Matching | | | | Variant Aggregation | | | |
|---|---|---|---|---|---|---|---|---|
| | Pick Coke Can | Move Near | Open/Close Drawer | Average | Pick Coke Can | Move Near | Open/Close Drawer | Average |
| RT-1 (begin) (Brohan et al., 2022) | 2.7% | 5.0% | 13.9% | 6.8% | 2.2% | 4.0% | 6.9% | 4.2% |
| RT-1 (15%) (Brohan et al., 2022) | 71.0% | 35.4% | 56.5% | 60.2% | 81.3% | 44.6% | 26.7% | 56.2% |
| RT-1 (converged) (Brohan et al., 2022) | **85.7%** | 44.2% | **73.0%** | 74.6% | 89.8% | 50.0% | 32.3% | 63.3% |
| HPT (Wang et al., 2024) | 56.0% | 60.0% | 24.0% | 46.0% | – | – | – | – |
| TraceVLA (Zheng et al., 2024) | 28.0% | 53.7% | 57.0% | 42.0% | 60.0% | 56.4% | 31.0% | 45.0% |
| RT-1-X (O'Neill et al., 2024) | 56.7% | 31.7% | 59.7% | 53.4% | 49.0% | 32.3% | 29.4% | 39.6% |
| RT-2-X (O'Neill et al., 2024) | 78.7% | 77.9% | 25.0% | 60.7% | 82.3% | 79.2% | **35.3%** | 64.3% |
| Octo-Base (Team et al., 2024) | 17.0% | 4.2% | 22.7% | 16.8% | 0.6% | 3.1% | 1.1% | 1.1% |
| OpenVLA (Kim et al., 2024) | 16.3% | 46.2% | 35.6% | 27.7% | 54.5% | 47.7% | 17.7% | 39.8% |
| RoboVLM (zero-shot) (Li et al., 2024b) | 72.7% | 66.3% | 26.8% | 56.3% | 68.3% | 56.0% | 8.5% | 46.3% |
| RoboVLM (fine-tuning) (Li et al., 2024b) | 77.3% | 61.7% | 43.5% | 63.4% | 75.6% | 60.0% | 10.6% | 51.3% |
| SpatialVLA (zero-shot) (Qu et al., 2025) | 81.7% | 85.7% | 56.0 % | 74.4% | 89.7% | 79.7% | 33.0% | 67.4% |
| SpatialVLA (fine-tuning) (Qu et al., 2025) | 82.5% | 85.7% | 54.6% | 74.3% | **90.6%** | 79.1% | 26.2% | 65.3% |
| *ReFineVLA* (Ours) | 83.0% | **95.3%** | 51.4% | **76.6%** | 83.8% | **88.1%** | 34.4% | **68.8%** |

**Overall Performance:** The results across both SimplerEnv benchmarks (Table 1 and Table 2) reveal that *ReFineVLA* achieves the highest average success rates in nearly all settings, reflecting improved learning behavior and generalization capacity. On the WidowX benchmark (Table 1), *ReFineVLA* achieves a 47.7% average success rate, outperforming the second-best SpatialVLA (42.7%) by 5.0%. It achieves near-perfect performance on the *Put Eggplant in Yellow Basket* task with 95.2% success, closely following SpatialVLA's 100.0%, while demonstrating substantial improvements in more complex spatial reasoning tasks, such as *Put Spoon on Towel* (21.4% over SpatialVLA) and *Put Carrot on Plate* (8.3%). These results indicate enhanced spatial awareness and manipulation capabilities enabled by reasoning-guided supervision.

In the SimplerEnv Google Robot tasks (Table 2), *ReFineVLA* achieves the highest average success in both *visual matching* (76.6%) and *variant aggregation* (68.8%) settings, outperforming SpatialVLA (74.3% and 65.3%) by 2.3% and 3.5%, respectively. Notably, *ReFineVLA* surpasses all baselines in the *Move Near* task with 95.3% accuracy (9.6% over SpatialVLA) and on *Open/Close Drawer* in variant aggregation with 34.4% (8.2%). Therefore, these experimental results highlight *ReFineVLA*'s robustness under spatial, visual, and temporal variations, thanks to its structured reasoning learning objective. For a comprehensive breakdown and further discussion of per-task results on SimplerEnv Google Robot benchmarks, please refer to Table 8 in Appendix C.2.

## 5.3 Ablation Studies

To analyze the performance of *ReFineVLA*, we further study the following key questions:

**1) How does reasoning supervision affect learning during fine-tuning and generalizing robotic tasks?** To further analyze how reasoning supervision enhances model performance, we conduct an ablation study using the SimplerEnv Google Robot benchmark (Table 3). We compare SpatialVLA fine-tuning against two variants of *ReFineVLA*: transfer fine-tuning only, and transfer fine-tuning with reasoning learning. Without reasoning, *ReFineVLA* already reaches 70.8% of accuracy in visual matching settings and 67.4% of accuracy in variant aggregation settings. However, once reasoning supervision is introduced, performance rises to 76.6% and 68.8%, respectively. The largest improvements are observed in high-level tasks requiring spatial and temporal abstraction, such as *Move Near* (9.6% over SpatialVLA) and *Open/Close Drawer* (8.2%). These findings support our design that targeted fine-tuning of upper layers and explicit reasoning loss promotes more structured and transferable policies, enhancing robustness under visual and embodiment shift.

**Table 3: The Effect of Reasoning Supervision on Generalization and Transferability:** The comparisons between SpatialVLA fine-tuning against two variants of *ReFineVLA* on SimplerEnv Google Robot tasks: (i) with transfer fine-tuning, and (ii) with transfer fine-tuning with explicit reasoning supervision. **Bold** indicates the best result, and underline denotes the second-best result per column.

| | Visual Matching | | | | Variant Aggregation | | | |
| Robotic Task
VLA Baseline | Pick
Coke Can | Move Near | Open/Close
Drawer | Average | Pick
Coke Can | Move Near | Open/Close
Drawer | Average |
|---|---|---|---|---|---|---|---|---|
| SpatialVLA (fine-tuning) (Qu et al., 2025) | 82.5% | 85.7% | **54.6%** | 74.3% | **90.6%** | 79.1% | 26.2% | 65.3% |
| *ReFineVLA* with (i) | 78.1% | 83.3% | 50.9% | 70.8% | 83.7% | 86.4% | 32.0% | 67.4% |
| *ReFineVLA* with (ii) | **83.0%** | **95.3%** | 51.4% | **76.6%** | 83.8% | **88.1%** | **34.4%** | **68.8%** |

In addition, to further investigate the role of explicit reasoning supervision in multimodal learning, we analyze the changes in training loss and action accuracy before and after introducing the *ReFineVLA* approach. As shown in Figure 4, the training L1 loss of the *ReFineVLA* model decreases more rapidly than the baseline model and maintains consistently lower values throughout the fine-tuning process. The improved loss trajectory suggests that explicitly guiding reasoning encourages the model to optimize its parameters more efficiently, potentially due to more structured gradients resulting from reasoning supervision. Similarly, Figure 4 demonstrates that the training action accuracy of *ReFineVLA* progressively increases and consistently surpasses the baseline, reflecting enhanced model generalization and a more precise alignment of learned representations with task requirements. These observations indicate that incorporating explicit multimodal reasoning supervision, as done in *ReFineVLA*, provides meaningful learning signals that guide the optimization process towards more accurate and stable multimodal policy representations.

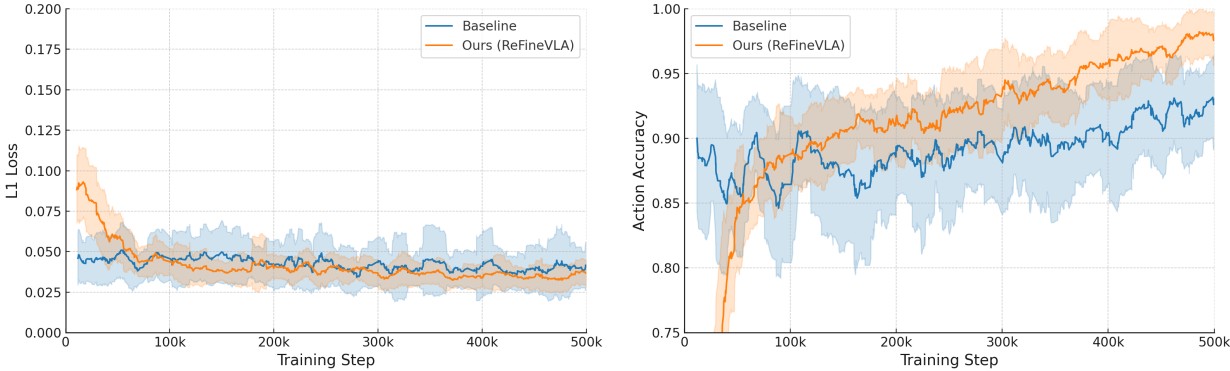

**Figure 4: Training Loss and Action Accuracy During Fine-tuning on Fractal Dataset**: Training progression of both the baseline and *ReFineVLA* models on the Fractal dataset: (*left*) the training L1 loss for ReFineVLA decreases more rapidly and remains lower throughout the fine-tuning process compared to the baseline, and (*right*) the action accuracy for *ReFineVLA* increases steadily and consistently surpasses that of the baseline at all training steps. The shaded regions represent the standard deviation across multiple runs. Together, these plots illustrate the differences in learning dynamics between the two approaches during fine-tuning.

**2) How does reasoning accuracy look like during *ReFineVLA* fine-tuning?** To better understand the internal effect of reasoning supervision in our proposed *ReFineVLA* model, we conduct an ablation analysis focused explicitly on reasoning accuracy throughout fine-tuning, as illustrated in Figure 5. Initially, reasoning accuracy improves rapidly during the early training steps, reflecting effective incorporation of reasoning-guided signals into the model's learning process. As training progresses, reasoning accuracy stabilizes, indicating that the model effectively maintains its reasoning capabilities over extended training periods. The observed stable trend and relatively low variance, depicted as the shaded

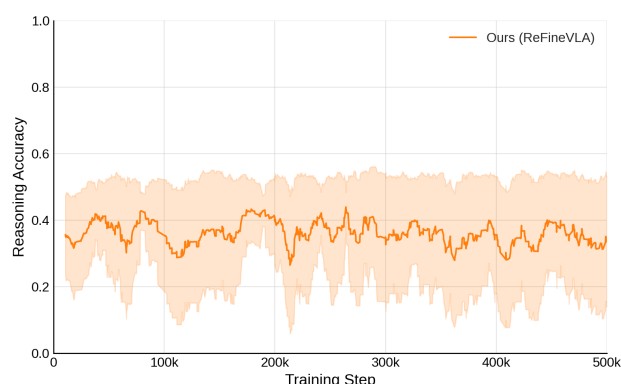

**Figure 5:** Reasoning accuracy over training steps.

regions, suggest robust internalization of reasoning processes. Thus, this analysis highlights that the explicit reasoning supervision from *ReFineVLA* successfully enhances the model's capability to reason about multimodal inputs, supporting more interpretable and structured decision-making.

**3) How does attention behavior change prior to and post *ReFineVLA* fine-tuning?** To understand how reasoning supervision impacts model behavior, we analyze attention maps before and after fine-tuning with *ReFineVLA*. As shown in Figure 6, before fine-tuning, VLA models tend to focus narrowly on immediate action targets – often overlooking contextual elements such as the spatial arrangement of objects or instruction – relevant cues. This behavior reflects the model's tendency to learn direct input-to-action mappings without deeper scene understanding.

After applying *ReFineVLA*, we observe a consistent shift in attention toward semantically meaningful regions, empirically revealing that the model learns to reason more holistically about the task by integrating visual and linguistic information over time. Figure 6 shows that the model's attention aligns better with the task instruction, supporting more robust and interpretable action decisions. These findings prove that *ReFineVLA* improves performance and promotes more structured, multimodal representations and human-understandable decision-making processes.

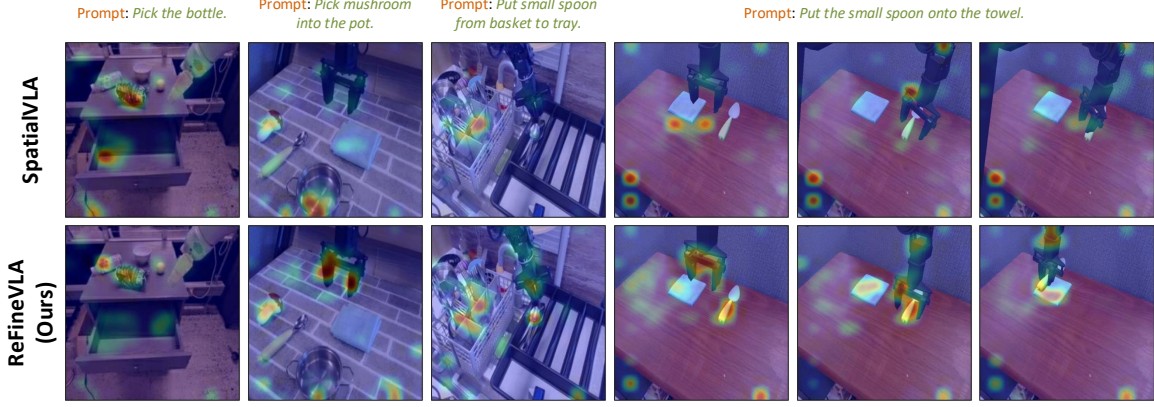

**Figure 6: Attention Visualization of *ReFineVLA* and SpatialVLA:** *RefineVLA* shows better attention to related entities within the given observations conditioned by the input prompts than what SpatialVLA does.

**4) How does *ReFineVLA* reason on complex or long-horizon robotic manipulation tasks?** Figure 7 illustrates a reasoning procedure of *ReFineVLA* for a robotic actuator actions alongside the step-by-step task planning. For complex tasks that require visual and instruction understanding, like *Close the drawer*, *ReFineVLA* shows a fine-grained instruction and corresponding actions to achieve the goal. *ReFineVLA*'s capability continues with long-horizon tasks, such as *Move the coke can near the orange*. The model can still generate proper actions with corresponding reasoning steps, not limited to observation, situation analysis, and spatial reasoning. It can be observed that *ReFineVLA* can identify the objects present in embodied

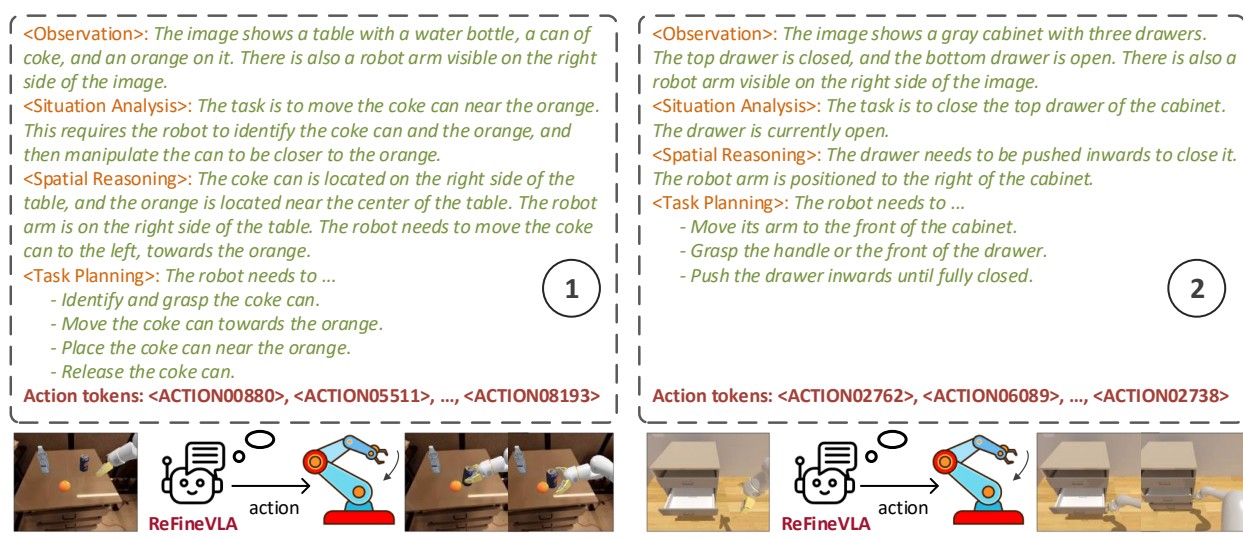

**Figure 7: Chain-of-Thought Reasoning of *ReFineVLA*:** *RefineVLA* shows step-by-step reasoning to accomplish the prompted task given the initial observation. The examples illustrate the queries (1) for placing the coke can near the orange in the tabletop setting and (2) for closing the drawer while it is opening. More examples are shown in Figure 9 in Appendix C.1.

scenes and estimate their approximate relative positions, which is critical for action learning. Furthermore, *ReFineVLA* is able to guide the robot to the next sub-actions, eventually achieving the goal task.

**5) How does *ReFineVLA* perform with different weighting of the reasoning loss, guided by the teacher's multimodal supervision?** An important hyperparameter in *ReFineVLA* is the reasoning loss weight $\lambda_r$, which determines the strength of the reasoning loss guided by the teacher's multimodal supervision. A higher $\lambda_r$ encourages the model to focus more on generating detailed step-by-step rationales, potentially improving task-level understanding and interpretability. However, when $\lambda_r$ is set too high, it leads to overemphasis on reasoning generation, resulting in noisy outputs or misaligned attention that distracts from critical visual features such as the robot end-effector or target object. In contrast, a smaller $\lambda_r$ reduces the risk of distraction but may produce shallow or underdeveloped reasoning traces, weakening the benefit of teacher-guided supervision. As shown on the left side of Figure 8, setting $\lambda_r = 0.3$ achieves the best performance, yielding a 9.7% improvement in task success, empirically showing that integrating teacher-provided reasoning can meaningfully enhance decision quality. However, excessively low and high values of $\lambda_r$ decrease performance, indicating the need to balance action prediction and reasoning learning.

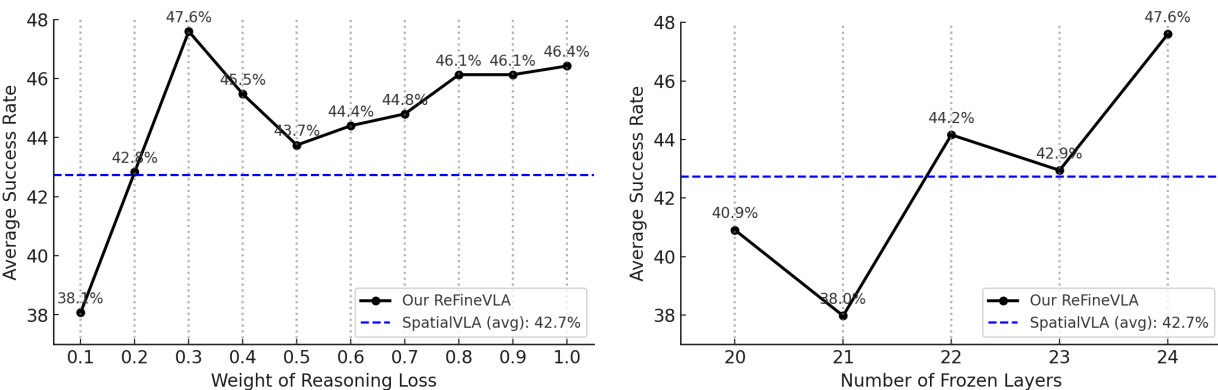

**Figure 8:** (*left*) Average success rate of *ReFineVLA* with different weights of reasoning loss $\lambda_r$ during teacher-guided fine-tuning for multimodal reasoning. (*right*) Average success rate of *ReFineVLA* with different numbers of frozen layers during teacher-guided fine-tuning for multimodal reasoning.

**6) How does *ReFineVLA* perform under different numbers of frozen layers during learning of teacher-guided multimodal reasoning supervision?** We find that freezing lower layers helps preserve general visual and linguistic representations learned from large-scale pre-training, while allowing upper layers to specialize for structured reasoning. The performance peaks when freezing the first 24 transformer layers, resulting in an 8.2% improvement of task success rate, as shown on the right side of Figure 8, which suggests that the upper layers of the model are most critical for absorbing high-level reasoning supervision, while the lower layers should retain pre-trained perceptual grounding. In contrast, freezing too few layers can potentially degrade these generalizable features, leading to noisy attention and less stable reasoning outputs. Freezing too many layers restricts the model's learning capacity and limits its ability to align with teacher-guided rationales.

For more ablation studies and results, please see Appendix C.

## 6    Conclusions and Discussions

In this work, we proposed *ReFineVLA*, a teacher-guided fine-tuning framework that enhances VLA models with explicit multimodal reasoning. By leveraging structured rationales from an expert reasoning teacher, *ReFineVLA* trains policies that jointly predict actions and generate step-by-step reasoning, enabling more profound understanding of complex and long-horizon robotic tasks. Through selective layer tuning and penalty on weights for reasoning-governed loss, our method preserves generalizable features while injecting high-level reasoning capabilities. Moreover, experiments across simulated and real-world robotic settings in SimplerEnv with WindowX Robot and Google Robot tasks demonstrate that *ReFineVLA* outperforms existing baselines in performance and interpretability, establishing a promising direction for reasoning-driven generalist VLA-based robot policies.

While *ReFineVLA* demonstrates strong performance and improved multimodal reasoning, several promising avenues remain for exploration. One direction is to scale reasoning supervision through human-in-the-loop refinement or self-improving teacher models using reinforcement learning. Besides that, extending this into real-world robotic systems will also help evaluate its robustness in sim-to-real transfer settings. Finally, incorporating memory mechanisms or temporal context could enable reasoning across long-horizon tasks.

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

# A    Code, Model, and Data Availability

To encourage future work in this research area, we provide public access to our codebase, models, and all Chain-of-Thought (CoT) reasoning datasets from our *ReFineVLA* framework.

▷ **Code Repository:**
  The source code for generating CoT reasoning datasets, model fine-tuning, evaluation, and attention visualization is available at: `ReFineVLA`

▷ **Pre-trained Models:**

  - WidowX (Table 1): `ReFineVLA_CoT_BrigeV2`.
  - Google Robot (Table 2): `ReFineVLA_CoT_Fractal`.
  - SpatialVLA Backbone: `spatialvla-4b-224-pt`.

▷ **CoT Datasets:**

  - Fractal_CoT: `Fractal_CoT`.
  - BrigeV2_CoT: `BrigeV2_CoT`.

▷ **Attention Visualization Scripts:**
  (`attention_vis/`) contains scripts for extracting and visualizing cross-modal attention maps from Re-FineVLA models, as used in our supplementary figures.

▷ **CoT Data Generation Scripts:**
  Scripts for generating CoT traces for Bridge-v2 and Fractal datasets are provided as:
  `./bridge_v2_add_task_reasoning`
  `./fractal_add_task_reasoning`

All resources are released for reproducibility and benchmarking under anonymous review. Detailed usage instructions are provided in `README.md`.

# B   More Implementation Details

For the VLA backbone, we employ SpatialVLA (Qu et al., 2025) (2B parameters, PaliGemma-2 VLM), pre-trained on Open X-Embodiment (O'Neill et al., 2023) and RHT20 (Fang et al., 2024) datasets. All model fine-tuning is performed on multi-node clusters with NVIDIA H100 GPUs, using our shell scripts.

## B.1   Automated Data Processing for Reasoning Trace Generation

To enable reasoning-augmented training, we construct a scalable pipeline to annotate every step of robot demonstration trajectories with structured multimodal reasoning traces. Our approach processes RLDS-format TFRecord datasets (*i.e.*, BridgeData-v2 and Fractal datasets) by extracting per-step multi-view image observations and the associated language instructions. For each step, we use a reasoning teacher model (*i.e.*, Gemini-2.0) to generate a detailed reasoning trace, encompassing visual observation, situational analysis, spatial reasoning, and explicit task planning. The resulting traces are saved in JSON format and are aligned to each episode and action step.

**Prompt Design:** We use a prompt to elicit detailed, step-by-step reasoning from the Gemini-2.0 teacher. For every set of images and instructions, the API is queried with:

```
You are a robot.  Given the images from different angles and instructions,
observe the scene and reason step-by-step about what you see, what each
object is, and what actions might be possible.
Instruction:  <language instruction> Now think carefully and describe:
#1 <Observation>:  What do you see in the image?
#2 <Situation Analysis>:  What is happening in the scene, and what is the
task?
#3 <Spatial Reasoning>:  How are the objects arranged, and what spatial
relationships matter for completing the task?
#4 <Task Planning>:  What are the logical steps to achieve the task, and
what should be the robot's next action?
```

The above prompt is to produce interpretable, chain-of-thought (CoT) reasoning traces. Example outputs and their impact on model performance are further illustrated below.

**Processing Workflow:** The pipeline is fully parallelizable (e.g., via Python's `concurrent.futures`) for efficient, large-scale data annotation, ensuring every demonstration frame is paired with a high-quality, multi-modal reasoning trace for use in ReFineVLA's reasoning-augmented training.

- **TFRecord Parsing:** Each RLDS TFRecord file is parsed to extract robot demonstration episodes, retrieving step-wise images and language instructions.
- **Multiview Image Aggregation:** At each time step, images from multiple camera viewpoints are aggregated as model input.
- **Teacher Model Query:** For every step, we send images and the corresponding instruction to the Gemini-2.0 API using the structured prompt above, obtaining a reasoning trace.
- **Trace Saving:** For each episode, the sequence of reasoning traces is saved as a JSON list, indexed by step, for downstream fine-tuning.

**Note:** The code release includes our full data processing scripts, including parallelized trace generation and integration with Gemini 2.0. For additional qualitative examples, see Table 4, Table 5, and Table 6.

## B.2   Training Details

We sample and annotate reasoning traces for approximately $125,000$ trajectories from BridgeData-v2 (Walke et al., 2023) and Fractal (Brohan et al., 2022) datasets using our multimodal reasoning teacher pipeline in Section 4. Fine-tuning on the reasoning-augmented dataset is conducted on one H100 node (2 GPUs) for one epoch, from 52 to 72 hours, depending on the dataset size.

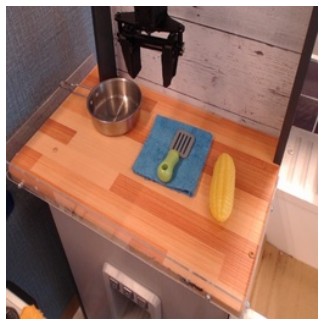

**Task Instructions Reasoning Prompt**: You are a robot. Given the image and the instruction to *Move the spatula to the bottom left side of the table*, scan the scene and reason step-by-step about what you see, what each object is, what the planning steps are for a robotic controller, and what actions might be possible? Think carefully and describe the steps followed the key questions:
#1 <Observation>: What do you see in the images?
#2 <Situation Analysis>: What is happening in the scene?
#3 <Spatial Reasoning>: How are the objects arranged, and what spatial relationships matter for completing the task?
#4 <Task Planning>: What are the logical steps to achieve the task, and what should be the robot's sequential actions?

**Task Instruction**: Move the spatula to the bottom left side of the table

**<Reasoning>**

**<Observation>**: I observe a toy kitchen setup with a wooden tabletop. On the tabletop, there is a silver pot, a blue cloth, a green and grey spatula, and a yellow toy corn cob.

**<Situation Analysis>**: The scene shows various toy kitchen items on a table. The task is to move the spatula to the bottom left side of the table.

**<Spatial Reasoning>**: The spatula is currently resting on the blue cloth, which is situated in the center-right of the table. The pot is on the left side of the table, and the corn cob is on the right. The "bottom left side of the table" is an empty area where the spatula should be moved.

**<Task Planning>**: The robot needs to pick up the spatula and place it in the designated bottom left area of the table: <logical_steps><1> Identify and locate the spatula <2> Move gripper to spatula <3> Grasp the spatula <4> Lift the spatula <5> Move gripper to target <6> Place the spatula <7> Release the spatula <sub_action> Move its arm to a position suitable for grasping the spatula.

**Table 4:** Instruction understanding with CoT reasoning for "*Move the spatula*" task.

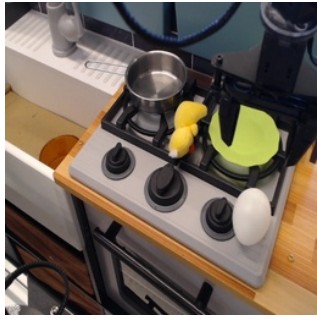

**Task Instructions Reasoning Prompt**: You are a robot. Given the image and the instruction to *put egg on plate and stuffed duck in pot or pan on stove*, scan the scene and reason step-by-step about what you see, what each object is, what the planning steps are for a robotic controller, and what actions might be possible? Think carefully and describe the steps followed the key questions:
#1 <Observation>: What do you see in the images?
#2 <Situation Analysis>: What is happening in the scene?
#3 <Spatial Reasoning>: How are the objects arranged, and what spatial relationships matter for completing the task?
#4 <Task Planning>: What are the logical steps to achieve the task, and what should be the robot's sequential actions?

**Task Instruction**: put egg on plate and stuff the duck in the pot or pan on the stove

**<Reasoning>**

**<Observation>**: I see a toy kitchen with a stove, sink, pot, green plate, yellow stuffed duck, white egg, and a robotic gripper arm.

**<Situation Analysis>**: The scene shows a toy kitchen with various items on the stovetop. The robotic arm is present and capable of interacting with the objects. The task requires two distinct actions: putting the egg on the plate and putting the stuffed duck in the pot.

**<Spatial Reasoning>**: All target objects and destinations are on the stove and within reach of the robotic arm. The egg is near the plate, and the stuffed duck is near the pot.

**<Task Planning>**: The robot needs to pick up the egg and place it on the plate, then pick up the stuffed duck and place it in the pot: <logical_steps><1> Pick up egg <2> Place egg on plate <3> Pick up stuffed duck <4> Place stuffed duck in pot <sub_action> The robot first attempts to put the egg on the plate.

**Table 5:** Instruction understanding with CoT reasoning for "*Put egg on plate and duck in pot*" task.

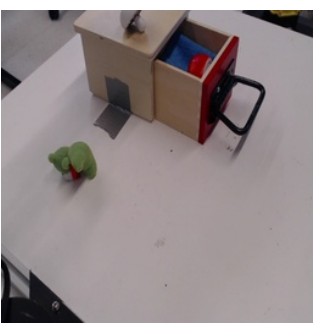 **Task Instructions Reasoning Prompt**: You are a robot. Given the image and the instruction to *Put the red brush on the table*, scan the scene and reason step-by-step about what you see, what each object is, what the planning steps are for a robotic controller, and what actions might be possible? Think carefully and describe the steps followed the key questions:
#1 <Observation>: What do you see in the images?
#2 <Situation Analysis>: What is happening in the scene?
#3 <Spatial Reasoning>: How are the objects arranged, and what spatial relationships matter for completing the task?
#4 <Task Planning>: What are the logical steps to achieve the task, and what should be the robot's sequential actions?

**Task Instruction**: Put the red brush on the table

**<Reasoning>**

**<Observation>**: I see a white table, a wooden box with an open drawer containing a blue cloth and a red brush, and a green object on the table.

**<Situation Analysis>**: The red brush is inside a confined drawer, requiring careful gripper access. The table offers ample clear space for placement.

**<Spatial Reasoning>**: The drawer constrains access to the brush, which may require a specific approach vector. The table has flat open space suitable for placing the brush.

**<Task Planning>**: <logical_steps> <1> Identify and locate red brush (in drawer) <2> Plan grasp to extract brush from drawer <3> Move gripper to brush and grasp <4> Lift brush from drawer <5> Identify and locate clear table space <6> Move brush to table and place <7> Release brush <sub_action> Reach into the drawer to grasp the red brush.

**Table 6:** Instruction understanding with CoT reasoning for "*Put the red brush on the table*" task.

## B.3 Fine-tuning Hyperparameters

We list our model's hyperparameters in the pre-training and fine-tuning stages, as summarized in Table 7.

| Hyperparameters | Fine-tuning Settings |
|---|---|
| GPUs | 2 |
| GPUs per Node | 2 |
| Per-Device Batch Size | 32 |
| Gradient Accumulation | 1 |
| Number of Workers | 1 |
| Epochs | 1 |
| Learning Rate | 2e-5 |
| Shuffle Buffer Size | 131072 |
| Image Resolution | $224 \times 224$ |
| **Action Token Size** | **12** |
| **Reasoning Token Size** | **244** |
| LoRA Params (`use_all_lora`, `lora_alpha`, etc.) | Off (default) |
| Vision Zoe Depth | On |
| FlashAttention v2 | On |
| Checkpoint Interval (`save_steps`) | 500 |

**Table 7:** Fine-tuning hyperparameters and settings for *ReFineVLA*.

## B.4 Layer Freezing and Efficient Adaptation

During ReFineVLA fine-tuning, the vision encoder and early transformer blocks are frozen to preserve general visual-linguistic representations, while upper layers and the policy head are updated to integrate reasoning efficiently. Freezing the first 24 transformer layers, as validated in our ablations (Figure 8), provides the best trade-off between stability and adaptation.

### B.5 Implementation of Forward Computation for Reasoning and Per-Step Reasoning Trace Generation

For reproducibility, we provide Algorithm 2 as a detailed pseudo-code of our forward computation for both reasoning and action losses, which is implemented in our codebase. Also, the pseudo-code for per-step reasoning trace generation using Gemini 2.0 API is provided in Algorithm 3.

---

**Algorithm 2:** Pseudo-code for *ReFineVLA*'s Forward Computation for Reasoning and Action Losses.

```
def forward(
    input_ids, pixel_values=None, intrinsic=None, attention_mask=None,
    position_ids=None, past_key_values=None, token_type_ids=None,
    cache_position=None, inputs_embeds=None, labels=None,
    use_cache=None, output_attentions=None, output_hidden_states=None,
    return_dict=None, num_logits_to_keep=0
):
    if inputs_embeds is None:
        inputs_embeds = embed_tokens(input_ids).clone()

    if use_spatial_token:
        spatial_selected = (input_ids >= action_token_begin_idx) & (
            input_ids < action_token_begin_idx + spatial_token_num
        )
        inputs_embeds[spatial_selected] = spatial_embed_tokens(
            input_ids[spatial_selected] - action_token_begin_idx
        )

    if pixel_values is not None:
        image_features = get_image_features(pixel_values, intrinsic)
        special_image_mask = (input_ids == image_token_index).unsqueeze(-1)
        inputs_embeds = inputs_embeds.masked_scatter(special_image_mask, image_features)

    causal_mask = update_causal_mask(
        attention_mask, token_type_ids, past_key_values,
        cache_position, input_ids, inputs_embeds
    )

    outputs = language_model(
        attention_mask=causal_mask, position_ids=position_ids,
        past_key_values=past_key_values, inputs_embeds=inputs_embeds,
        use_cache=use_cache, output_attentions=output_attentions,
        output_hidden_states=output_hidden_states, return_dict=return_dict,
        cache_position=cache_position, num_logits_to_keep=num_logits_to_keep,
    )

    logits = outputs.logits

    if labels is not None:
        shift_logits = logits[..., :-1, :]
        shift_labels = labels[..., 1:]

        mask_action_tokens = (shift_labels >= translation_token_start_idx) & (
            shift_labels <= gripper_token_end_idx
        )

        shift_logits_action = shift_logits[mask_action_tokens].contiguous()
        shift_labels_action = shift_labels[mask_action_tokens].contiguous()
        shift_logits_reason = shift_logits[~mask_action_tokens].contiguous()
        shift_labels_reason = shift_labels[~mask_action_tokens].contiguous()

        loss_fct = nn.CrossEntropyLoss()

        loss_action = loss_fct(
            shift_logits_action.view(-1, vocab_size),
            shift_labels_action.view(-1)
        )

        loss_reasoning = loss_fct(
            shift_logits_reason.view(-1, vocab_size),
            shift_labels_reason.view(-1)
        )

        loss = loss_action + lambda_r * loss_reasoning

    return loss, logits, outputs.hidden_states, outputs.attentions
```

---

**Algorithm 3:** Pseudo-code for Per-Step Reasoning Trace Generation with Gemini 2.0 API

```python
def generate_reasoning_traces(tfrecord_file):
    dataset = load_tfrecord(tfrecord_file)
    for episode_id, episode in enumerate(dataset):
        steps = extract_steps(episode)
        language_instruction = steps["language_instruction"]
        all_traces = []
        for t in range(num_steps(steps)):
            # Aggregate multi-view images at step t
            img_list = [steps[f"image_{i}"][t] for i in range(num_cameras)]
            # Query Gemini API for reasoning using the Figure 9 prompt
            trace = call_gemini_api(img_list, language_instruction)
            all_traces.append({t: trace})
        save_json(all_traces, f"reasoning/{file_id}_{episode_id}.json")

def call_gemini_api(img_list, language_instruction):
    contents = []
    for img in img_list:
        # Wrap image bytes for API
        type_part = types.Part.from_bytes(
            data=img, mime_type='image/png',
        )
        contents.append(type_part)
    # Append structured reasoning prompt (see Figure 9)
    contents.append(f'''
        You are a robot.
        Given the images from different angles and instruction,
        observe the scene and reason step-by-step about what you see,
        what each object is, and what actions might be possible.
        Instruction: {language_instruction}
        Now think carefully and describe:
        #1 <Observation>: What do you see in the image?
        #1 <Situation Analysis>: What is happening in the scene and what is the task?
        #1 <Spatial Reasoning>: How are the objects arranged
        and what spatial relationships matter for completing the task?
        #1 <Task Planning>: What are the logical steps to achieve the task,
        and what should be the robot's next action?
    ''')
    try:
        response = gemini_client.models.generate_content(
            model='gemini-2.0-flash',
            contents=contents
        )
        return response.text
    except Exception as e:
        print(f"Error API GEMINI: {e}")
        return ""
```

## B.6 Implementation of VLA Attention Visualization

To support the analyses in Section 3, we outline our pipeline for extracting and aggregating attention maps from VLA models. During inference, we enable attention outputs to capture the attention weights for action and reasoning tokens across all layers and attention heads. These attention maps are then fused using a top-$k$ aggregation strategy to highlight the most salient input regions that influence the model's predictions. By reshaping and normalizing these fused maps into interpretable grids, we visualize model focus as heatmaps overlaid on the input images. All visualizations in Section 3 are generated using the following implementation. Note that Algorithm 4 elaborates the extraction and top-$k$ aggregation of cross-modal attention for target action tokens.

---

**Algorithm 4:** Pseudo-code for Attention Extraction and top-$k$ Aggregation in VLA Models

```python
def extract_and_fuse_attention(model, inputs, target_token_indices, top_k=5):
    # Run inference with output_attentions enabled
    outputs = model.generate(
        **inputs,
        output_attentions=True,
        return_dict_in_generate=True,
    )
    attentions = outputs["attentions"]  # (layers, heads, src_len, tgt_len)

    # Collect attention maps for each target token (e.g., action or reasoning token)
    attn_maps = []
    for t_idx in target_token_indices:
        # Gather attention weights for this token across all layers and heads
        layer_head_attns = [
            attn[:, :, :, t_idx] for attn in attentions  # attn: (batch, heads, src, tgt)
        ]  # list of (batch, heads, src_len) per layer
        layer_head_attns = torch.stack(layer_head_attns)  # (layers, batch, heads, src_len)

        # Aggregate the top-k values (across layers and heads) using the max or mean strategy
        fused = topk_aggregate(layer_head_attns, k=top_k, method="max")  # (src_len,)
        attn_maps.append(fused)
    return attn_maps  # List of attention maps, one per token

def topk_aggregate(attn_tensor, k=5, method="max"):
    # attn_tensor: (layers, batch, heads, src_len)
    attn_tensor = attn_tensor.flatten(0, 2)  # (layers*heads, batch, src_len)
    topk_vals, _ = torch.topk(attn_tensor, k=k, dim=0)  # (k, batch, src_len)
    if method == "max":
        return topk_vals.max(dim=0).values.squeeze(0)
    elif method == "mean":
        return topk_vals.mean(dim=0).squeeze(0)
    else:
        raise ValueError("Unsupported fusion method.")
```

---

# C  Additional Ablation Studies

## C.1  Reasoning-Grounded, Multi-Modal Actions by *ReFineVLA*

To demonstrate *ReFineVLA*'s ability to generate multi-modal and reasoning-compliant actions, we present qualitative results from challenging real-world-like table-top and drawer manipulation scenarios.

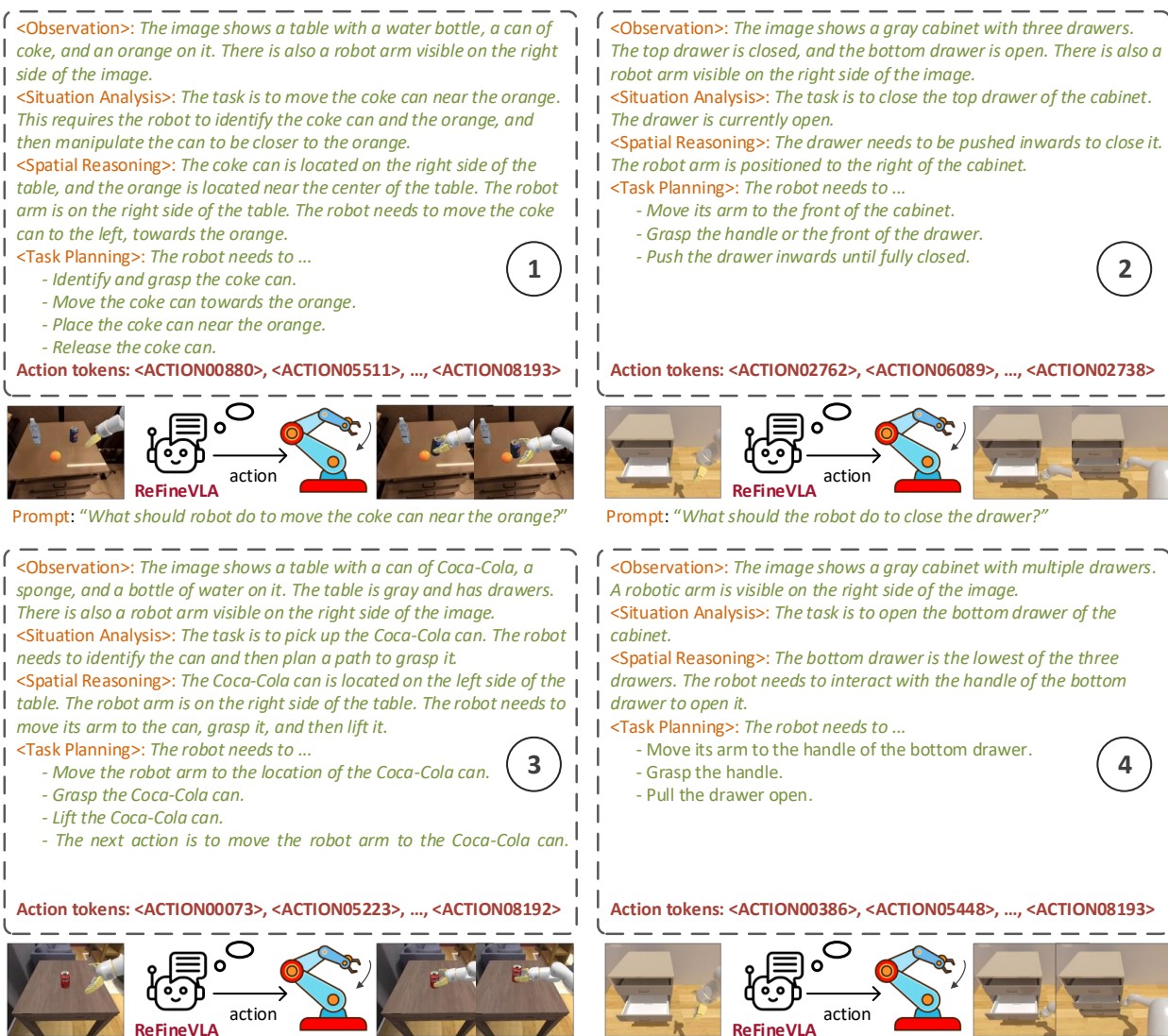

**Figure 9: Chain-of-Thought Reasoning of *ReFineVLA*:** *RefineVLA* shows step-by-step reasoning with corresponding actions to accomplish the prompted task given the initial observations and linguistic command. The extended illustrations of Figure 7 are provided.

Figure 9 depicts step-by-step examples where *ReFineVLA* observes the environment through multi-view images, grounds its actions in explicit multi-stage reasoning, and generates a sequence of action tokens that follow the reasoning trace. Each panel shows *ReFineVLA* first analyzing the visual scene (*Observation*), interpreting the current task (*Situation Analysis*), performing spatial and physical reasoning about object locations and relationships (*Spatial Reasoning*), and then generating a concrete action plan (*Task Planning*). For instance, in the "*move the coke can near the orange*" task (Panel 1), *ReFineVLA* decomposes the high-level goal into subtasks: identifying, grasping, moving, and releasing the can. Similarly, in drawer manipulation tasks (Panel 2 and Panel 4), the model reasons about the cabinet state and the spatial layout, plans to approach and operate the correct handle, and executes the whole action sequence accordingly.

*ReFineVLA*'s explicit reasoning structure allows for diverse, context-dependent solutions. For grasping and placement tasks (Panel 1 and Panel 3), the model flexibly chooses different grasp approaches and trajectories based on the current spatial arrangement, demonstrating multi-modal action generation. The underlying action token sequence adapts to the inferred plan and spatial cues, showing that the model does not collapse to a single, rigid policy but samples actions probabilistically and with high fidelity to the reasoning trace. This multi-modal reasoning ability is a direct consequence of the joint fine-tuning on both action and reasoning supervision, which leads the model to fit action targets and the causal logic and spatial semantics necessary for robust decision-making. Empirically, this approach yields more accurate grasps, improved success rates on sequential tasks, and lower action prediction error compared to unimodal or non-reasoning baselines.

Furthermore, our analysis confirms that *ReFineVLA*'s generated actions are tightly coupled to its internal reasoning trace. Even when the model's reasoning contains mistakes (for example, misidentifying an object or spatial relationship), the subsequent action sequence reliably follows from the reasoning output, demonstrating unified, reasoning-compliant behavior. For example, if *ReFineVLA* incorrectly reasons which drawer to open, it will still faithfully execute the corresponding drawer action described in its plan, between reasoning and action execution. These findings suggest that continued improvement is enhanced by reasoning accuracy, through precise annotation, richer multimodal context, or stronger vision-language models.

In summary, we conclude that *ReFineVLA* is able to produce multi-modal, context-sensitive actions grounded in explicit, interpretable reasoning traces, consistently aligning its execution with its internal thought process. This structured, principled, reasoning-aligned control framework provides new opportunities for analysis and improvement in generalist robotic policy learning.

**Table 8:** Evaluation results across different policies on SimplerEnv with Google Robot tasks in two settings: visual aggregation and visual matching. The evaluated tasks are "*Pick Coke can*", "*Move Near*", and "*Open/Close Drawer*" at different configurations. The overall performance across the tasks is also reported in the last column.

| | | Pick Coke Can | | | | Move Near | Open/Close Drawer | | | Overall |
| | Robotic Task | | | | | | | | | Average |
| VLA Baseline | | Horizontal Laying | Vertical Laying | Standing | Average | Average | Open | Close | Average | Average |
|---|---|---|---|---|---|---|---|---|---|---|
| Variant Aggregation | RT-1 (begin) | 2.2% | 1.3% | 3.1% | 2.2% | 4.0% | 0.5% | 13.2% | 6.9% | 4.2% |
| | RT-1 (15%) | 92.0% | 70.4% | 81.3% | 81.3% | 44.6% | 21.2% | 32.3% | 26.7% | 56.2% |
| | RT-1 (converged) | 96.9% | 76.0% | 96.4% | 89.8% | 50.0% | 27.0% | 37.6% | 32.3% | 63.3% |
| | TraceVLA | — | — | — | 60.0% | 56.4% | — | — | 31.0% | 45.0% |
| | RT-1-X | 56.9% | 20.4% | 69.8% | 49.0% | 32.3% | 6.9% | 51.9% | 29.4% | 39.6% |
| | RT-2-X | 82.2% | 75.4% | 89.3% | 82.3% | 79.2% | 33.3% | 37.2% | 35.3% | 64.3% |
| | Octo-Base | 0.5% | 0.0% | 1.3% | 0.6% | 3.1% | 0.0% | 2.1% | 1.1% | 1.1% |
| | OpenVLA | 71.1% | 27.1% | 65.3% | 54.5% | 47.7% | 15.8% | 19.5% | 17.7% | 39.8% |
| | RoboVLM (zero-shot) | 77.8% | 48.0% | 79.1% | 68.3% | 56.0% | 1.6% | 15.3% | 8.5% | 46.3% |
| | RoboVLM (fine-tuning) | 93.8% | 49.8% | 83.1% | 75.6% | 60.0% | 2.6% | 18.5% | 10.6% | 51.3% |
| | SpatialVLA (zero-shot) | 93.1% | 83.6% | 92.6% | 89.7% | 79.7% | 19.0% | 47.0% | 33.0% | 67.4% |
| | SpatialVLA (fine-tuning) | 95.7% | 82.0% | 94.2% | 90.6% | 79.1% | 18.5% | 33.8% | 26.2% | 65.3% |
| | ***ReFineVLA* (fine-tuning)** | 78.8% | 80.4% | 92.1% | 83.8% | 88.1% | 18.6% | 50.3% | 34.4% | 68.8% |
| Visual Matching | RT-1 (Begin) | 5.0% | 0.0% | 3.0% | 2.7% | 5.0% | 0.0% | 27.8% | 13.9% | 6.8% |
| | RT-1 (15%) | 86.0% | 79.0% | 48.0% | 71.0% | 35.4% | 46.3% | 66.7% | 56.5% | 60.2% |
| | RT-1 (Converged) | 96.0% | 90.0% | 71.0% | 85.7% | 44.2% | 60.1% | 86.1% | 73.0% | 74.6% |
| | HPT | — | — | — | 56.0% | 60.0% | — | — | 24.0% | 46.0% |
| | TraceVLA | — | — | — | 28.0% | 53.7% | — | — | 57.0% | 42.0% |
| | RT-1-X | 82.0% | 33.0% | 55.0% | 56.7% | 31.7% | 29.6% | 89.1% | 59.7% | 53.4% |
| | RT-2-X | 74.0% | 74.0% | 88.0% | 78.7% | 77.9% | 15.7% | 34.3% | 25.0% | 60.7% |
| | Octo-Base | 21.0% | 21.0% | 9.0% | 17.0% | 4.2% | 00.9% | 44.4% | 22.7% | 16.8% |
| | OpenVLA | 27.0% | 3.0% | 19.0% | 16.3% | 46.2% | 19.4% | 51.8% | 35.6% | 27.7% |
| | RoboVLM (zero-shot) | 85.0% | 43.0% | 90.0% | 72.7% | 66.3% | 28.7% | 25.0% | 26.8% | 56.3% |
| | RoboVLM (fine-tuning) | 94.0% | 47.0% | 91.0% | 77.3% | 61.7% | 33.3% | 53.1% | 43.5% | 63.4% |
| | SpatialVLA (zero-shot) | 69.0% | 79.7% | 96.4% | 81.7% | 85.7% | 49.1% | 62.9% | 56.0% | 74.4% |
| | SpatialVLA (fine-tuning) | 84.5% | 67.8% | 95.2% | 82.5% | 85.7% | 45.3% | 63.8% | 54.6% | 74.3% |
| | ***ReFineVLA* (fine-tuning)** | 79.8% | 75.0% | 94.1% | 83.0% | 95.3% | 43.5% | 59.3% | 51.4% | 76.6% |

### C.2 Details of Evaluation on SimplerEnv with Google Robot Tasks

Table 8 presents comprehensive results across all tasks for the SimplerEnv Google Robot evaluation. Our *ReFineVLA* is directly compared against strong baselines, including RT-1 (in various training stages), RT-2-X, SpatialVLA, and others, in both **Variant Aggregation** and **Visual Matching** settings.

**Variant Aggregation:** In the "*Pick Coke Can*" task, *ReFineVLA* achieves competitive results: for can-standing configuration, *ReFineVLA* achieves 92.1%, just below the top RT-1 (converged) at 96.4% and SpatialVLA at 94.2%, surpassing all other baselines. Meanwhile, at more difficult "Vertical Laying" and "Horizontal Laying" task configurations, *ReFineVLA* scores 80.4% and 78.8%, respectively, all ranking in the top three. Notably, for the "*Move Near*" task, *ReFineVLA* delivers the **best performance overall** (88.1%), outperforming all prior baselines, followed by RT-2-X (79.2%) and SpatialVLA (79.1%). In "*Open/Close Drawer*" tasks, *ReFineVLA* leads on "*Close Drawer*" with 50.3%, and maintains comparable results for "*Open Drawer*" of 18.6%, outperforming fine-tuned SpatialVLA and many other methods on average. As a result, *ReFineVLA* achieves the **highest overall average** (68.8%).

**Visual Matching:** For "*Pick Coke Can*" in can-standing configuration, *ReFineVLA* achieves a success rate of 94.1%, nearly matching those performances of SpatialVLA and RT-1 (converged). In "*Move Near*" task, *ReFineVLA* achieves the **highest score of all models (95.3%)**. Meanwhile, for "*Close/Open Drawer*", *ReFineVLA* also ranks in the top performance with 51.4% and 59.3%, respectively. Overall, *ReFineVLA* achieves the **best average success rate of** 76.6%, outperforming both fine-tuned SpatialVLA (74.3%) and all policy learning or VLM baselines, showing superior generalization and robustness in visually-diverse scenarios.

In general, *ReFineVLA* is consistently top-2 or top-3 across nearly every sub-task and outperforms all prior methods in aggregate averages for both evaluation settings. Its task-level robustness, particularly excelling in "*Move Near*", "*Pick Coke Can*", and "*Close Drawer*" tasks, demonstrates the advantages of explicit reasoning-augmented training for robust, generalist robotic policies.

