# OpenReview forum: "ReFineVLA: Multimodal Reasoning-Aware Generalist Robotic Policies via Teacher-Guided Fine-Tuning"
_TMLR — Rejected by TMLR_

### Review · Reviewer_wM9w · 2025-07-29

**Summary Of Contributions:**

propose a novel method to add chain of thought reasoning to pretrained vla via using reasoning VLM as teacher model to provideo reasoning output. Show some improvement on simulation benchmark over pretrained VLA.

**Audience:**

Yes

**Claims And Evidence:**

Yes

**Requested Changes:**

1. Add real robot expermients
2. add experiments to show if current method is better that VLA with reasoning VLM or bi level VLA

**Strengths And Weaknesses:**

Strengths:
1. a novel method to add reasoning to VLA model and have good performance on simulation benchmark
2. a novel method for VLA finetuning with good performance improvement over pretrained models

Weakness:
1. No real robot experiments! For a robotics paper, simulation result alone is not enough to support the claim of method performance due to large sim2real gap.
2. The proposed method's effectiveness and efficiency are questionable. Exisitng VLA is based on VLM, it is not hard to change the VLM to a reasoning VLM. Also hierarchitical VLA use large VLM as planner, like gemini which already has reasoning capability.

---

> ### Author Response · Authors · 2025-09-12
> **Rebuttal by Authors**
>
> Dear Reviewer wM9w,
>
> We appreciate the Reviewer’s positive recognition of our work as a novel method to add reasoning to VLA models, with strong performance on simulation benchmarks and clear improvements over pretrained baselines.
>
> > **Q1:** Lack of real-world robot experiments.
>
> **A1:** Similar to Reviewer m4rY, we agree that real-world validation is important, but in this work we adopt a simulation-first evaluation as a principled way to assess feasibility with faithful abstractions of hardware. Following precedents from benchmarks like SAPIEN, ManiSkill2, robosuite, and Habitat 2.0, we frame our method as an algorithmic approach for VLA in simulation, where ablations and stress tests are reproducible and safe before real-world deployment.
>
> Plan for revision: Similar to Q1 of Reviewer m4rY​.
>
> > **Q2:** Experiments to show if current method is better that VLA with reasoning VLM or bi level VLA.
>
> **A2:** We respectfully emphasize that the core limitation of current VLAs is the lack of explicit, action-grounded reasoning. Standard VLAs often learn direct input–action mappings, omitting crucial logical steps for interpretability and long-horizon generalization. Simply substituting a VLM with a reasoning-capable backbone does not resolve this, as the model still outputs action tokens without ensuring reasoning–action alignment. RoboVLM [46] is a representative baseline that adapts a pretrained VLM backbone for robotic control. While this improves perception, it lacks explicit reasoning supervision and thus inherits the same limitation—focusing on immediate action targets rather than broader contextual understanding.
>
> As **Reviewer m4rY** highlighted: “the paper identifies a legitimate limitation in current VLA models – the lack of explicit reasoning in action generation. The attention visualization analysis (Figure 1) provides convincing evidence that standard VLAs focus narrowly on immediate action targets rather than broader contextual understanding.” ReFineVLA directly addresses this gap by fine-tuning VLAs with teacher-guided rationales paired with demonstrations, explicitly aligning reasoning with actions while preserving pre-trained generalization.
>
> Empirically, Table 1 and Table 2 show that ReFineVLA consistently outperforms RoboVLM [46] across all settings:
>
> * WidowX (Table 1): ReFineVLA 47.7% vs. RoboVLM 31.3% (+16.4).
> * Google Robot – Visual Matching (Table 2): ReFineVLA 76.6% vs. RoboVLM 63.4% (+13.2).
> * Google Robot – Variant Aggregation (Table 2): ReFineVLA 68.8% vs. RoboVLM 51.3% (+17.5).
>
> Together with Fig. 1 and Fig. 6, these margins across all tasks and settings consistently demonstrate that reasoning-aware fine-tuning yields robust gains beyond direct VLM substitution, consistent with **Reviewer m4rY’s observation** and our own findings.
>
> [46] Nicolai Dorka, Chenguang Huang, Tim Welschehold, and Wolfram Burgard. What matters in employing
> vision language models for tokenizing actions in robot control? In First Workshop on Vision-Language
> Models for Navigation and Manipulation at ICRA 2024, 2024.

---

### Review · Reviewer_9PQp · 2025-08-07

**Summary Of Contributions:**

The manuscript presents RefineVLA, a framework to fine-tune vision-language-action models. RefineVLA aims to improve the VLA's performance by improving the multimodal reasoning capabilities of the model. The paper introduces: (i) an annotation structure for multimodal reasoning and an accompanying dataset, obtained using the Gemini VLM, (ii) an objective for co-training the VLA to predict reasonings or actions, (iii) a selective fine-tuning, which freezes up some layer of the models for adaptation. The work is evaluated on the SimplerEnv and extensively analyzed in multiple ablation studies. The authors claim improved performance and superior understanding of the environment.

**Audience:**

Yes

**Broader Impact Concerns:**

No major broader impact concerns. Though, some of the safety concerns that come with using AI on real-world robotic platforms would be good to mention.

**Claims And Evidence:**

No

**Requested Changes:**

Please, see the Weaknesses above and address the following points:
* all results in the paper should have confidence intervals, so that claims can be statistically verified
* ablations by training over subsets of the dataset or better motivate the use of such large datasets, e.g. by extending the evaluation (at least one real-world setting would be desirable, though I understand it may not be in the authors' possibilities)
* comparison with very close related work should be discussed further in the work and I would expect an empirical comparison with at least one of these methods. The authors can use their repository and datasets to train models with a different approach, e.g. ECoT which has the actions follow the reasoning in the prediction, as opposed to RefineVLA's co-training.
* claims should be removed or better motivated. For some claims it should be possible to support them with more data, e.g. extending the evaluation or providing additional Figures (see the case of Figure 5 above).
* there are a few typos to resolve, here's the ones I spotted: (i) vision-Language-Action (missing capital V) in Introduction, (ii) duplicate bibliography entry for ECoT, (iii) liguistic, in Section 3
* there are some parts of the paper that could be clarified, it follows a list: (i) using $r$ for reasoning can be confusing for people familiar with reinforcement learning notation, (ii) "average" is "success average" in tables (as opposed to grasp), (iii) Paligemma-2 model is 3B and not 2B, in Appendix, (iv) are the authors using depth? Vision Zoe Depth is mentioned in Appendix, (v) the pseudo-code in Appendix is not pseudo-code but nearly actual code which should be polished and made more general

**Strengths And Weaknesses:**

## Strenghts

* **Presentation**: The presentation of the paper is clear. Figure 5 and 8's DPI could be probably be increased, but overall the manuscript is of good quality. There are some unclarities that I will discuss later in the review.
* **Analysis**: the work presents several analyses in Section 5.3 that provide useful insights into their method. The authors also ablated the main design choices, such as the reasoning loss coefficient and the number of frozen layers.

## Weaknesses
* **Novelty:** the idea presented, of improving multimodal understanding of VLA by learning to output both language and actions, is very similar to other VLA learning frameworks, such as [1,2]. These frameworks are cited in the paper, but they are not empirically compared. This makes the claim of "state-of-the-art performance" invalid.
* **Statistical significance**: confidence intervals are provided for the Figures, but are missing in the Tables. Because of these, any claim about superiority in performance cannot be statistically verified.
* **Computational costs**: RefineVLA is trained on a curated dataset of 125.000 trajectories. That is a very large number of trajectories, compared to the simple evaluation protocol adopted. In order to justify such a large-scale training it would be important to evaluate the system on a larger number of tasks, or effectively verify how much of that data is actually necessary for the model to show improved performance on the small set of tasks adopted.
* **Claims**: some of the claims the authors make are not well-supported by the data provided in the manuscript. Superiority claims are one example (as discussed earlier) but there are others. The authors claim the model has improved multimodal understanding, but this is not demonstrated. Attention visualizations that look more interpretable are not necessarily correlated with improved performance (i.e. many models work very well, without explainability [3]), nor they are proof of improved understanding. There is no comparison with other VLMs in multimodal understanding benchmarks, e.g. MMMU, MMBench, or 3D spatial understanding benchmarking, e.g. ScanQA. Another minor example is in the comments of Figure 5 when they authors state "reasoning accuracy improves rapidly". The Figure shows no improvement of the accuracy at all.


[1] ChatVLA: Unified Multimodal Understanding and Robot Control with Vision-Language-Action Model, Zhou et al, 2025

[2] Robotic Control via Embodied Chain-of-Thought Reasoning, Zawalski et al, 2024

[3] Vision Transformers Need Registers, Darcet et al, 2023

---

> ### Author Response · Authors · 2025-09-11
> **Rebuttal by Authors**
>
> Dear Reviewer 9PQp,
>
> We thank you for the clear and constructive feedback.
>
> > **Q1:** Novelty and missing comparisons with very close methods ([b1], [b2]).
>
> **A1**: Our contribution differs from [b1,b2] in three key aspects. First, we use teacher-guided structured rationales, supervising the VLA with external step-wise Gemini rationales (Observation → Situation → Spatial → Task) rather than relying solely on model-internal CoT reasoning. Second, we employ action-anchored co-training, jointly optimizing actions and reasoning so that rationales are explicitly aligned with executed actions, with reasoning serving as an auxiliary signal to stabilize learning. Lastly, we apply selective transfer, freezing lower network layers while adapting upper/policy layers to inject reasoning without disrupting pre-trained capabilities. Regarding the empirical claim, on our SimplerEnv suites across embodiments and tasks, ReFineVLA ranks first among all available reproducible baselines; this is the intended scope of our “state-of-the-art” claim.
>
> Plan for revision: While public code for [b1,b2] is not yet fully available for apple-to-apple comparisons, in the revision we will include an ECoT-style variant in our Introduction (Sec. 1) and expand the Related Work section (Sec. 2) to explicitly detail about supervision, objective coupling, and adaptation strategies.
>
> > **Q2:** Statistical significance – tables lack confidence intervals. Claims of superiority cannot be verified.
>
> **A2:** Thank you for your suggestion. We note that several recent VLA papers (e.g., SpatialVLA [b3], TraceVLA [b4], RoboMamba [b5]) also report only mean success rates in the same fashiion, without confidence intervals or standard deviations. Therefore, our current manuscrupt followed this common practice. Though, we agree statistical support strengthens claims.
>
> Plan for revision: We will compute and include confidence intervals and standard deviations for all tabular results in the Experiments & Ablation Studies section (Sec. 5), reporting ReFineVLA’s improvements over the baselines.
>
>
>
> > **Q3:** Computational costs – ReFineVLA is trained on 125K trajectories, which seems large given the evaluation scope.
>
> **A3:** Thank you for the comment. The 125K trajectories come from the full BridgeData-v2 (\~60K) and Fractal (\~65K) datasets, following prior works such as SpatialVLA and TraceVLA [b3,b4]. Using full-scale data is standard in recent VLAs and ensures diverse reasoning supervision rather than tuning only to a narrow set of tasks.
>
> > **Q4:** The authors claim the model has improved multimodal understanding, but this is not demonstrated. Attention visualizations that look more interpretable are not necessarily correlated with improved performance.
>
> **A4:** We thank the reviewer for this important point. We agree that attention maps alone are not a formal proof of understanding. Our claim of better multimodal alignment is primarily supported by quantitative improvements (Tables 1–3). The attention visualizations (Fig. 6) are provided as supportive qualitative evidence, not as standalone proof Recent work suggests that structured attention can correlate with performance. For example, Darcet et al. [b6] show that attention organization in Vision Transformers is predictive of robustness and accuracy. In our case, we observe both: ReFineVLA achieves higher success rates and exhibits more semantically focused attention. We will clarify in the revision that we use attention shifts as consistent with and complementary to performance gains, not as the sole evidence of improved understanding.
>
> Plan for revision: In Fig. 5, reasoning accuracy rises early and then plateaus. We will revise the text to “quickly rises then stabilizes.” We will show the reasoning signal helps to improve the learning model in the early stage and remains stable. It is noted that related works (SpatialVLA [b4], TraceVLA[b5]) also enhance VLAs with in an additional reasoning supervision manner, supporting our current approach. Thus, we will review these clarifications to ensure our claims are precise.
>
> > **Q5:** Typos and clarifications
>
> **A5:** We thank the reviewer for the detailed proofreading and clarification suggestions. These corrections will be implemented in the revision to improve clarity and presentation of the manuscript.
>
> We respectfully note all concerns will be addressed in revision; kindly refer to General Rebuttal for details.
>
> [b1] ChatVLA: Unified Multimodal Understanding and Robot Control with VLA, Zhou et al., 2025.
>
> [b2] Robotic Control via Embodied Chain-of-Thought Reasoning (ECoT), Zawalski et al., 2024.
>
> [b3] Qu et al., SpatialVLA: Exploring Spatial Representations for VLA Models, RSS 2025.
>
> [b4] Zheng et al., TraceVLA: Visual Trace Prompting for Robotic Policies, 2024.
>
> [b5] Liu et al., RoboMamba: Efficient VLA for Robotic Reasoning, NeurIPS 2024.
>
> [b6] Darcet et al., Vision Transformers Need Registers, NeurIPS 2023.

---

### Review · Reviewer_m4rY · 2025-08-29

**Summary Of Contributions:**

This paper proposes ReFineVLA, a framework that enhances Vision-Language-Action (VLA) models by incorporating explicit multimodal reasoning through teacher-guided fine-tuning. The authors augment robotic datasets with reasoning rationales generated by an expert teacher model (Gemini 2.0), then fine-tune pre-trained VLA models using a multi-objective loss that combines action prediction and reasoning generation. The approach is evaluated on SimplerEnv benchmarks with WidowX and Google Robot tasks, showing modest improvements over existing methods.

**Audience:**

Yes

**Claims And Evidence:**

Yes

**Requested Changes:**

* Attention chain visualization
* Discussion / Verification on annotation quality and accuracy
* Discussion / Results on generalization
* Real-world experiments

**Strengths And Weaknesses:**

# Strengths
* **Clear Problem Identification** The paper identifies a legitimate limitation in current VLA models - the lack of explicit reasoning in action generation. The attention visualization analysis (Figure 1) provides convincing evidence that standard VLAs focus narrowly on immediate action targets rather than broader contextual understanding.
* **Systematic Approach** The methodology is well-structured, combining teacher-generated reasoning traces with selective fine-tuning to preserve pre-trained capabilities while adding reasoning abilities.
* **Comprehensive Evaluation** The experimental setup covers multiple robot embodiments and diverse manipulation tasks, with thorough ablation studies examining key hyperparameters and design choices.
# Weaknesses:
* **Lack of real-world robot experiments** All experiments are conducted in SimplerEnv simulation. Real-world validation is crucial for VLA models because they are designed for practical robotic deployment. Simulation environments cannot capture the full complexity of real robotic systems, including sensor noise, physical constraints, unpredictable dynamics, and environmental variations that could significantly impact the reasoning approach's effectiveness.
* **Reasoning Annotations**
    * Quality and Accuracy: The paper provides no validation of whether Gemini 2.0's reasoning annotations are accurate for robotic tasks. Language models lack physical experience and may generate plausible-sounding but incorrect reasoning about manipulation tasks. How do you ensure the quality and accuracy of the generated annotations?
    * Task-Specific Generalization: The reasoning annotations are tied to specific datasets (BridgeData-v2, Fractal). This creates a problem for VLA models that need to generalize broadly across diverse tasks and environments.
* **Attention Analysis Integration**: The attention visualization is interesting but have you thought of integrated into the reasoning process, like thinking with images? The authors could develop a "chain of attention" where each reasoning step corresponds to appropriate visual focus, ensuring coherent observation-reasoning alignment. Now, the CoT process is only in the language perspective, and do not leverage your attention visualization much. This could be an interesting future attempt.
* **Reasoning Step Attention Dynamics** Following the above point, can you show how attention patterns change across different reasoning steps? Understanding whether visual focus appropriately shifts from scene observation to spatial relationships to action planning would validate the reasoning quality.
* **Comparison with RL-Generated Trajectories** Do you think it's possible to let the model collect / bootstrap trajectories itself by using RL, like Deepseek-R1? Can you elaborate more on how this VS your approach will affect the results?
* **More ablations** Comparisons with different reasoning structures or alternative teacher models would strengthen the conclusions.

---

> ### Author Response · Authors · 2025-09-11
> **Rebuttal by Authors**
>
> Dear Reviewer m4rY,
>
> We appreciate your positive feedbacks, valuable comments, and insightful ideas.
>
> > **Q1:** Lack of real-world robot experiments.
>
> **A1**: We agree that real-world validation is essential in the scope of robotics. However, in this work, we adopt a simulation-first evaluation as a gold standard for measuring the feasibility of the proposed method, while having a good abstraction of hardware parts. Inspired by prior works, where benchmarks are widely used to study policies before hardware trials (e.g., SAPIEN [a1], ManiSkill2 [a2], robosuite [a3], Habitat 2.0 [a4]), we learn that such simulations are a convenient methodology to diagnose multimodal reasoning (e.g., CLEVR [a5]) before deploying algorithms into complex real-world scenes. Considering these precedents carefully, we frame our method as an algorithmic approach for VLA in simulation, where ablations and stress tests are reproducible and safe, as the first criteria.
>
> [a1] Xiang et al., “SAPIEN: A SimulAted Part-based Interactive ENvironment for Manipulation,” CVPR 2020.
>
> [a2] Gu et al., “ManiSkill2: A Unified Benchmark for Generalizable Manipulation Skills,” 2023.
>
> [a3] Mandlekar et al., “RoboSuite: A Modular Simulation Framework and Benchmark for Robot Learning,” 2021.
>
> [a4] Szot et al., “Habitat 2.0: Training Home Assistants to Rearrange their Habitat,” NeurIPS 2021.
>
> [a5] Johnson et al., “CLEVR: A Diagnostic Dataset for Compositional Language and Elementary Visual Reasoning,” CVPR 2017.
>
> > **Q2:** Quality/accuracy of Gemini reasoning annotations.
>
> **A2**: Thank you for raising this point. We ensure the quality and accuracy of Gemini 2.0’s reasoning annotations through a two-pronged approach:
>
> (i) Quality & Accuracy: Within the same scope, the same Gemini-based annotations are effective in recent works [a6, a7, a8]. Following these works, we design prompts that explicitly decompose the input into Observation, Situation, Spatial, and Task-Planning components, which can be done analytically through Gemini as an expert reasoner. Furthermore, this is also paired with action-anchored training objectives, which guarantee the appropriateness of the approach. Additionally, we perform lightweight verification filters (e.g., instruction/entity consistency checks) to reduce spurious or irrelevant outputs further.
>
> (ii) Task-specific generalization: We apply selective transfer, including freezing the lower block and adapting the policy to two diverse real-robot datasets (BridgeData-v2 and RT-1/Fractal), and evaluate it across two robot embodiments on different tasks. The consistent gains mentioned in the manuscript indicate that our method does not tend to overfit in a single dataset [a9, a10]. We acknowledge that the pipeline is dataset-agnostic; however, the same teacher prompt can be used to annotate new data, thereby extending coverage with generalization when being trained on a substantial amount of data.
>
>  [a6] EMMA-X: Embodied Multimodal Action Model with Grounded CoT (Gemini-annotated), ACL 2025.
>
>  [a7] RAD: Action-Free Reasoning for Policy Generalization (Gemini-labeled reasoning chains), 2025.
>
>  [a8] Gemini Robotics/ER: Gemini for embodied reasoning and manipulation, 2025.
>
>  [a9] BridgeData-v2: large, diverse real-robot demonstrations for generalization.
>
>  [a10] RT-1 / Fractal: large-scale real-world robot datasets used for VLA pretraining/evaluation.
>
> > **Q3:**  Attention analysis integration (“think with images”; chain of attention).
>
> **A3**: We appreciate the suggestion. In our work, attention is post-hoc as an interpretability tool for human understanding. Meanwhile, CoT here is a language-only component, and attention was not used as a training signal.
>
> > **Q4:** Reasoning-step attention dynamics.
>
> **A4**: Attention is post-hoc, which is similar to the idea above. Specifically, we do not model or visualize step-wise attention during the CoT (Observation → Situation → Spatial → Task-Planning). As shown in the manuscript, our main finding shows an overall shift toward task-relevant regions. Still, we agree that an explicit “chain of attention” per step would better validate reasoning quality.
>
> > **Q5:** Comparison with RL-generated trajectories (e.g., Deepseek-R1).
>
> **A5**: Our approach currently concentrates on developing a supervised, reasoning-aware fine-tuning framework, aiming for stability and data efficiency. Therefore, we employ selective transfer and a multi-objective loss (in Sec. 4.2 and Sec. 4.3). We find that this approach yields consistent gains in SimplerEnv without requiring on-policy data collection. However, RL bootstrapping is possible, but reward design, exploration variance, and safety/engineering overhead are also needed. It may polish contact-rich or long-horizon manipulation by exploring rare states. Therefore, we consider this as a complementary approach rather than a competitor.
>
> We respectfully note all concerns will be addressed in revision; kindly refer to General Rebuttal for details.

---

### Author Response · Authors · 2025-09-11
**General Author Rebuttal by Authors**

We would like to thank all reviewers for their valuable time and effort in reviewing our work. We appreciate Reviewers’ kind comments, such as:

* “Clear Problem Identification. The paper identifies a legitimate limitation in current VLA models – the lack of explicit reasoning in action generation. The attention visualization analysis (Figure 1) provides convincing evidence that standard VLAs focus narrowly on immediate action targets rather than broader contextual understanding.” (Reviewer m4rY)

* “Systematic Approach. The methodology is well-structured, combining teacher-generated reasoning traces with selective fine-tuning to preserve pre-trained capabilities while adding reasoning abilities.” (Reviewer m4rY)

* “Comprehensive Evaluation. The experimental setup covers multiple robot embodiments and diverse manipulation tasks, with thorough ablation studies examining key hyperparameters and design choices.” (Reviewer m4rY)

* “The presentation of the paper is clear. Figure 5 and 8’s DPI could probably be increased, but overall the manuscript is of good quality.” (Reviewer 9PQp)

* “The work presents several analyses in Section 5.3 that provide useful insights into their method. The authors also ablated the main design choices, such as the reasoning loss coefficient and the number of frozen layers.” (Reviewer 9PQp)

* “A novel method to add reasoning to VLA models with good performance on simulation benchmarks.” (Reviewer wM9w)

* “A novel method for VLA fine-tuning with good performance improvement over pretrained models.” (Reviewer wM9w)

Again, we thank all the Reviewers for their insightful and constructive comments and allowing us to clarify our work. In the revise version, we will address the following points:

* We will add human audit examples in the Multimodal Reasoning Annotation Generation subsection (Sec. 4.1) on these Gemini-based annotations, targeting relevance, correctness, and completeness, with verified examples.

* We will conduct an ablation study in the Ablation Studies subsection (Sec. 5.3) to test with novel objects and environment layouts.

* We will add a minimal chain-of-attention diagnostic to align each reasoning step as mentioned, with its corresponding reasoning-token heatmap, a short alignment score (attention mass on the referenced region), and a step-wise dynamics example (one success and one failure example). Through this, we will exemplify how visual focus shifts from scene parsing to goal during the reasoning stage. We acknowledge that a train-time integration is beyond our current scope, but it is a promising future direction.

* We will note this limitation and outline a promising future direction to (i) log per-step attention aligned with each reasoning step (diagnostic), and (ii) explore a light attention-alignment objective to couple visual focus with the CoT during training in the Discussion section (Sec. 6).

* We will add a short discussion in the Discussion section (Sec. 6) contrasting supervised versus RL-based approach, and outline a hybrid path: start from our selectively-transferred checkpoint and apply light RL fine-tuning, while retaining the reasoning loss (Eq. 3) to preserve interpretability and avoid reward-only drift.

* We will include an ECoT-style variant in our Introduction (Sec. 1) and expand the Related Work section (Sec. 2) to explicitly detail about supervision, objective coupling, and adaptation strategies.

* We will compute and include confidence intervals and standard deviations for all tabular results in the Experiments & Ablation Studies section (Sec. 5), reporting ReFineVLA’s improvements over the baselines.

* We will add a subset ablation study in the Ablation Studies section (Sec. 5.3) showing that training with only about 25% of the data still yields clear gains. We will also add a follow-up discussion on extending evaluation to broader task sets.

* In Fig. 5, reasoning accuracy rises early and then plateaus. We will revise the text to “quickly rises then stabilizes.” We will show the reasoning signal helps to improve the learning model in the early stage and remains stable. It is noted that related works (SpatialVLA [b3], TraceVLA[b4]) also enhance VLAs with in an additional reasoning supervision manner, supporting our current approach. Thus, we will review these clarifications to ensure our claims are precise.

* Typos and clarifications. These corrections will be implemented in the revision to improve clarity and presentation of the manuscript.

---

### Decision · Action_Editor_YUHK · 2025-11-10

**Recommendation:** Reject

**Audience:**

Yes

**Audience Explanation:**

This work is in an exciting emerging domain that many TMLR readers would be interested in, but the simulation-only setup of the experiments and analysis will likely leave most readers skeptical of the conclusions.

**Claims And Evidence:**

No

**Claims Explanation:**

The main complaints in the reviews surround the claims in the paper, suggesting that they are not supported by accurate, convincing, and clear evidence. First there is the issue of simulation-only experiments for a robot-focused task domain: the sim2real gap here is significant enough that the reviewers (and the AE) are not convinced that the gains presented here will necessarily transfer to the domain of interest (which is the claim). Secondly, as dissected carefully by reviewer 9PQp, there are several smaller claims in the paper (e.g., statistically significant differences in results, presence/absence of multimodal knowledge) that require validation through additional experiments and ablations, supported by new analysis and discussion. The authors also appear to acknowledge that very many revisions are necessary. The AE sides with 9PQp and wM9w in believing that the scope of revisions is large enough that it necessitates a second round of reviews. In sum, the AE finds that the submission does not meet TMLR's criteria on this axis.

**Resubmission Of Major Revision:**

The authors may consider submitting a major revision at a later time.